# CERSA: Cumulative Energy-Retaining Subspace Adaptation for Memory-Efficient Fine-Tuning

## Abstract

To mitigate the memory constraints associated with fine-tuning large pre-trained models, existing parameter-efficient fine-tuning (PEFT) methods, such as LoRA, rely on low-rank updates. However, such updates fail to fully capture the rank characteristics of the weight modifications observed in full-parameter fine-tuning, resulting in a performance gap. Furthermore, LoRA and other existing PEFT methods still require substantial memory to store the full set of frozen weights, limiting their efficiency in resource-constrained settings. To address these limitations, we introduce **Cumulative Energy-Retaining Subspace Adaptation (CERSA)**, a novel fine-tuning paradigm that leverages singular value decomposition (SVD) to retain only the principal components responsible for 90% to 95% of the spectral energy. By fine-tuning low-rank representations derived from this principal subspace, CERSA significantly reduces memory consumption. We conduct extensive evaluations of CERSA across models of varying scales and domains, including image recognition, text-to-image generation, and natural language understanding. Empirical results demonstrate that CERSA consistently outperforms state-of-the-art PEFT methods while achieving substantially lower memory requirements. The code will be released.

## 1 Introduction

Fine-tuning pre-trained large models for specific tasks has become a common practice to achieve superior performance in both natural language processing and computer vision domains (Hu et al., 2022; Sun et al., 2024; Meng et al., 2024). Pre-trained models, which have been trained on extensive and diverse datasets (Deng et al., 2009; Lin et al., 2014), accumulate rich and general knowledge, enabling them to outperform models trained from scratch. However, fine-tuning the entire pre-trained model typically demands substantial computational resources like memory, particularly for large-scale models based on transformer architectures, such as ViT-Large (Dosovitskiy, 2021) and DeBERTaV3 (He et al., 2023). Unlike massive training clusters, often equipped with thousands of GPUs for pre-training, fine-tuning is more likely to occur on consumer-grade GPUs to support diverse downstream applications. Consequently, reducing the number of tunable parameters and the memory footprint has become a focal point in parameter-efficient fine-tuning (PEFT) research (Hu et al., 2022; Zi et al., 2023; Zhang et al., 2023a; Kopiczko et al., 2024; Gu et al., 2022; Ren et al., 2024; Valipour et al., 2023).

Existing PEFT methods aim to fine-tune only a small subset of parameters within pre-trained models (Rebuffi et al., 2017; Li & Liang, 2021; Lester et al., 2021), which significantly reduces memory requirements. Since fewer parameters are updated during backpropagation, the demand for memory to store gradients and optimizer states decreases. Among the most popular methods are LoRA (Hu et al., 2022) and its variants (Zi et al., 2023; Zhang et al., 2023a; Kopiczko et al., 2024; Ren et al., 2024; Sun et al., 2024; Meng et al., 2024), which introduce two low-rank matrices, $B \in \mathbb{R}^{m \times r}$ and $A \in \mathbb{R}^{r \times n}$ ($r \ll m, r \ll n$), to reparameterize fine-tuning as $B \times A$. Here, the pre-trained weight matrix $W \in \mathbb{R}^{m \times n}$ is frozen, and only the newly added low-rank matrices are trained.

Despite these advances, most existing methods focus on reducing memory usage by exploiting the low-rank nature of gradients during training (Hu et al., 2022; Zi et al., 2023; Kopiczko et al., 2024; Gu et al., 2022; Ren et al., 2024; Valipour et al., 2023). However, the full weight matrices must be stored in memory, with few approaches directly compressing the pre-trained weights. As a result, the total memory consumption for weights, gradients, and optimizer states often remains tied to the size of the

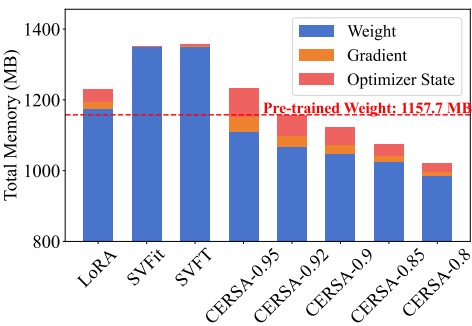 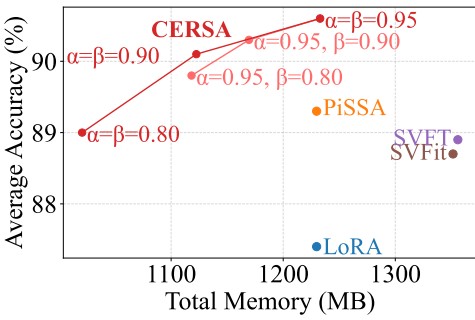

Figure 1: Memory footprint comparison for fine-tuning ViT-Large (Dosovitskiy, 2021).

Figure 2: Average accuracy(see Tab. 7) versus total memory usage on ViT-Large (Dosovitskiy, 2021)

pre-trained weights. Besides, SVFit (Sun et al., 2024) and SVFT (Lingam et al., 2024) use singular value decomposition (SVD) to compress pre-trained weights but require storing two full singular vector matrices of size $\mathbb{R}^{n \times n}$, limiting the memory savings despite their low trainable parameter counts (see §2.2). Furthermore, recent studies (Shuttleworth et al., 2024) reveal a key limitation of LoRA: it introduces intruder dimensions that degrade the model's performance on learned tasks. These findings motivate us to directly preserve the major components of pre-trained weights, enabling memory-efficient fine-tuning while maintaining the prior knowledge encoded during pre-training.

To this end, we propose **Cumulative Energy-Retaining Subspace Adaptation (CERSA)**, a memory-efficient fine-tuning method for pre-trained weights. The key idea is to apply SVD to each weight matrix and truncate it to retain only the components that preserve most of the cumulative energy (typically 90%–95%). Since singular values of weight matrices follow a heavy-tailed distribution, a small set of dominant singular vectors suffices for adapting the model to downstream tasks. As illustrated in Fig. 3, depending on the matrix position, retaining only 10%–50% of the original dimensions is often enough to capture the principal energy. This enables substantial memory savings during fine-tuning with minimal performance loss. For example, in ViT-Large (Dosovitskiy, 2021), keeping 95% of the cumulative energy yields a memory footprint comparable to LoRA (rank=32) (Hu et al., 2022), as shown in Fig. 1. Reducing the threshold to 90% further lowers memory usage below that of the original pre-trained weights, while causing only a negligible drop—about 0.3% on average across three image classification datasets (Tab. 4). As illustrated in Fig. 2, CERSA achieves a clearly superior accuracy-memory trade-off compared to baseline methods, making it especially effective under strict memory constraints.

The primary contributions of this paper are as follows:

- We propose CERSA, a memory-efficient PEFT method that uses SVD to retain the primary cumulative energy of pre-trained model weights and fine-tunes within the principal subspace. This reduces memory usage below the weight size, improves fine-tuning efficiency compared to LoRA (Hu et al., 2022), and minimizes the forgetting of prior knowledge.

- We provide a theoretical analysis of CERSA, showing that fine-tuning within the principal cumulative energy subspace is sufficient for adapting the model to downstream tasks. This subspace overlaps significantly with those required for most tasks, helping retain pre-trained knowledge during fine-tuning.

- We comprehensively evaluate CERSA on image classification and natural language understanding tasks. Results demonstrate that CERSA consistently outperforms state-of-the-art PEFT baselines while achieving the best accuracy-memory trade-off, highlighting its effectiveness under constrained memory budgets.

## 2 RELATED WORK

### 2.1 LOW-RANK ADAPTATION

LoRA (Hu et al., 2022) is a key method in PEFT, reducing memory usage by decomposing weight updates into low-rank matrices while keeping pre-trained weights frozen. This enables the efficient fine-tuning of large models. Enhancements to LoRA (Hu et al., 2022) can be categorized into three types: weight-driven, data-driven, and adaptive methods.

Weight-driven methods add adapters derived from weight decomposition on top of frozen pre-trained weights, directly manipulating the weight space via matrix decompositions and orthonormal constraints. Representative approaches, including PiSSA (Meng et al., 2024), OLoRA (Wang et al., 2023), MiLoRA (Wang et al., 2024a), LoRA-XS (Bałazy et al., 2024), and DoRA (Liu et al., 2024), introduce techniques such as SVD-based initialization, QR-based orthonormal initialization, and minor singular component adaptation to enhance representation learning and convergence speed.

Data-driven methods leverage model activations, gradients, or data distributions to guide adapter updates. Techniques like LoRA-GA (Wang et al., 2024b), LoRA-Pro (Wang et al., 2024c), LaMDA (Azizi et al., 2024), and EVA (Paischer et al., 2024) employ strategies such as aligning low-rank gradients with full fine-tuning gradients and performing SVD on mini-batch activations for variance-aware initialization, thereby improving adaptation efficiency through data-informed adjustments.

Adaptive methods dynamically configure adapters by task characteristics or layer importance to optimize parameter utilization. Approaches such as AdaLoRA (Zhang et al., 2023b) and EVA (Paischer et al., 2024) employ rank allocation by layer importance and variance-aware adjustments, effectively balancing model capacity with computational cost to achieve efficient fine-tuning.

Despite these advancements, most LoRA-based methods store the entire frozen weight matrix with multiple adapters, offering limited memory savings over the original LoRA (Hu et al., 2022). This underscores the need for more efficient methods to further reduce memory and computational costs.

## 2.2 WEIGHT-DECOMPOSITION-BASED METHOD

To further minimize the number of parameters required for fine-tuning and reduce computational costs, weight-decomposition-based methods have been developed to process pre-trained weights. Generally, the basic step of weight-decomposition-based methods (Han et al., 2023) is to decompose the original weight matrix $W$ into $U$, $\Sigma$, and $V$. SVFit (Sun et al., 2024) fine-tunes only the top-$k$ singular values, freezing $U$ and $V$ to retain principal components. SVFT (Lingam et al., 2024) freezes $\Sigma$ and introduces a sparse adapter for task-specific adaptation. SVDiff (Han et al., 2023) applies singular value fine-tuning to diffusion models, reducing storage while mitigating overfitting. WeLore (Jaiswal et al., 2024) optimizes rank reduction across layers by identifying low-rank components for selective fine-tuning, enhancing efficiency with minimal performance loss.

Although these methods reduce trainable parameters, they require storing $U$ and $V$, doubling the original weight size (Lingam et al., 2024). When accounting for gradients and optimizer states, their memory footprint exceeds twice that of the pre-trained weights, making them more memory-intensive than LoRA (Hu et al., 2022) and other PEFT methods.

## 3 METHODOLOGY

Fine-tuning pre-trained models using Singular Value Decomposition (SVD) has proven to be an effective approach for adapting large-scale models while minimizing parameter updates (Han et al., 2023; Sun et al., 2024; Lingam et al., 2024). However, traditional SVD-based fine-tuning incurs substantial computational and memory overhead by necessitating the storage of two full decomposed matrices, effectively doubling memory consumption compared to standard weight storage. Moreover, freezing the left and right singular matrices restricts the model's expressiveness, making it suboptimal relative to full-parameter fine-tuning. To address these limitations, we propose a constrained optimization framework that selectively updates the principal components using trainable matrices while discarding components associated with minor singular vectors. By fine-tuning within the principal subspace of the weight matrix, our method retains the core representational capacity of the pre-trained model while significantly reducing memory requirements, thereby enabling efficient and stable adaptation to downstream tasks.

### 3.1 LAYER-WISE RANK SELECTION

In existing methods, a persistent challenge is that, regardless of the attached adapter, the original pre-trained weights $W \in \mathbb{R}^{m \times n}$ will impose a memory cost of $\mathcal{O}(mn)$ and incur a computational overhead of $\mathcal{O}(mn)$ during forward propagation, even for fully frozen matrices. As a result, no matter how parameter-efficient the fine-tuning method is, this storage and computation burden remains

Figure 3: Preserved singular value indices in ViT-Large (Dosovitskiy, 2021) (pre-trained on ImageNet-21K (Deng et al., 2009)) across layers and weight matrices under different cumulative energy retention rates. The query (**Q**), key (**K**), value (**V**), and projection (**P**) matrices correspond to weight matrices in self-attention, while up (**UP**) and down (**DN**) matrices represent the weight matrices of the first and second linear operations of the multilayer perceptron (MLP), respectively.

unavoidable. Inspired by PiSSA (Meng et al., 2024), we propose to retain the most significant cumulative energy (Jolliffe, 2002) of the weight matrix in terms of the $U$ and $V$ matrices of its SVD, assuming that the subspace defined by $U$ and $V$ matrices is sufficient for most fine-tuning scenarios. To further reduce the dimensionality, we propose using truncated SVD as it provides the optimal low-rank approximation in terms of the Frobenius norm (Eckart & Young, 1936).

Moreover, the singular value distributions of pre-trained weights vary significantly across layers, influenced by both the layer depth and the type of weight matrix. To effectively extract the principal components across layers, we propose to retain the cumulative energy of the truncated SVD using the *cumulative energy retention rate* (Eckart & Young, 1936). This rate measures the proportion of total cumulative energy retained in the selected components after truncation and is calculated as:

$$\alpha = \frac{\sum_{i=1}^{k} s_i^2}{\sum_{j=1}^{N} s_j^2}, \tag{1}$$

where $s_i$ represents the singular values corresponding to the $i$-th principal component, $k$ denotes the number of selected singular values after truncation, and $N$ is the total number of singular values. The numerator, $\sum_{i=1}^{k} s_i^2$, represents the energy retained in the first $k$ singular values, corresponding to the $k$ most significant components of the matrix. The denominator, $\sum_{j=1}^{N} s_j^2$, represents the total energy of the original matrix. Consequently, $\alpha$ quantifies the ratio of retained to total energy, reflecting the proportion of the matrix's variance preserved in the truncated representation.

Eq. (1) implies that higher singular values contribute more to the total cumulative energy of the matrix. By setting a specific cumulative energy retention rate (*e.g.*, $\alpha = 0.95$) across different layers, one can determine the minimum rank $k$ required to retain a desired proportion of the cumulative energy of the weight matrix. Hence, the cumulative energy retention rate enables us to balance dimensionality reduction and information preservation.

Fig. 3 illustrates the preserved singular value indices in ViT-Large (Dosovitskiy, 2021), pre-trained on ImageNet-21K (Deng et al., 2009), after SVD decomposition. The rank values are computed across different layer types at various cumulative energy retention rates {0.8, 0.85, 0.9, 0.92, 0.95}. As observed, the indices of the preserved singular value arrays exhibit an increasing trend from the lower to upper layers, indicating that lower layers allow for greater compression. Additionally, compared to the MLP layers, the query, key, value, and projection matrices in the self-attention module have lower cutoff index values at the same cumulative energy retention rate. These observations underscore the importance of layer-wise rank selection in optimizing model efficiency.

### 3.2 TRAINABLE MATRIX IN THE PRINCIPAL SUBSPACE

By computing the cumulative energy, we establish the criterion for top-$k$ truncation of weight matrices. Using the truncation index in Fig. 3, we maximize the removal of residual ranks layer-wise, thereby optimizing memory usage. The retained ranks determined by the chosen cumulative energy threshold represent the trade-off between performance and memory budget.

For a descending sequence of $N$ singular values: $\sigma_1^2 \geq \sigma_2^2 \geq \cdots \geq \sigma_N^2$, we define two hyperparameters, $\alpha$ and $\beta$, to determine the preserved and trainable subspaces. The threshold $\alpha$ defines the retained subspace, while $\beta$ specifies the trainable portion within that subspace. $k_\alpha$ and $k_\beta$ are the smallest indices such that the cumulative sum reaches the proportions $\alpha$ and $\beta$, respectively:

$$k_\alpha = \min\left\{ k \;\middle|\; \frac{\sum_{i=1}^{k} \sigma_i^2}{\sum_{i=1}^{N} \sigma_i^2} \geq \alpha \right\}, \qquad k_\beta = \min\left\{ k \;\middle|\; \frac{\sum_{i=1}^{k} \sigma_i^2}{\sum_{i=1}^{N} \sigma_i^2} \geq \beta \right\}. \tag{2}$$

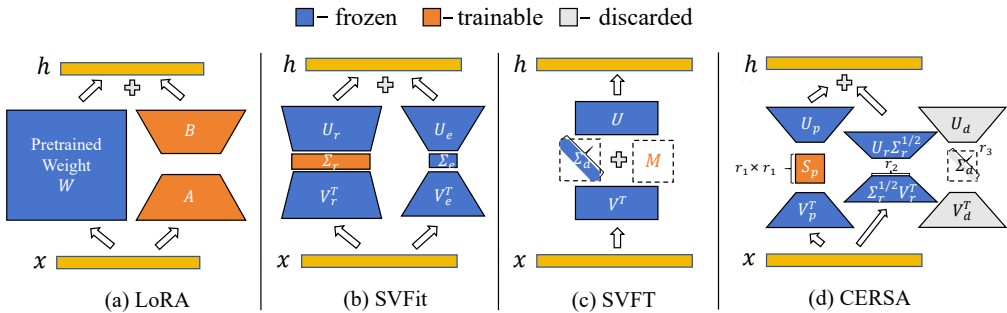

Figure 4: Comparison among LoRA (Hu et al., 2022), SVFit (Sun et al., 2024), SVFT (Lingam et al., 2024) and CERSA. (a) LoRA (Hu et al., 2022) uses two low-rank matrices to approximate weight updates during fine-tuning. (b) SVFit (Sun et al., 2024) initializes low-rank matrices through SVD of $W$ and trains only the most significant singular values as a vector. (c) SVFT (Lingam et al., 2024) freezes singular vectors while sparsely fine-tuning singular values. (d) CERSA discards redundant SVD components and only trains a core matrix initialized with the most significant singular values.

These two thresholds, $\alpha$ and $\beta$, divide the matrix into three distinct regions, as illustrated in Fig. 4(d). ❶ **Discarded Component:** We perform SVD on the pre-trained weights $W$, *i.e.*, $W = USV^T$. By setting the cumulative energy retention rate $\alpha$, the least important components are discarded by truncating the least $r_3 = N - k_\alpha$ significant singular vectors in $U$ and $V$. Unlike PiSSA (Meng et al., 2024) and SVFit (Sun et al., 2024), which retain the residual part by freezing it, CERSA eliminates redundant high-rank components that contribute only 5%-10% to the cumulative energy but occupying 50%-90% of the embedding dimensions, as determined by the SVD cumulative energy truncation index (Fig. 3). Most of the singular values in this discarded portion are near zero, indicating feature dimensions that are insignificant in the pre-trained weights. Despite that this is still a lossy compression, these high-rank components may not be well-aligned or optimally parameterized for downstream fine-tuning tasks. ❷ **Frozen Component**: Next, we introduce another hyperparameter, $\beta$, as a threshold to compute the rank $r_2 = k_\alpha - k_\beta$, which determines which of the remaining principal components will be frozen. The value of $\beta$ reflects the trade-off between preserving more pre-trained knowledge and utilizing additional dimensions to learn the feature distribution of downstream tasks for stronger fitting capability. In the image classification and text sequence classification tasks, we set $\beta = \alpha$, allowing all principal dimensions to participate in fine-tuning for maximum fitting performance. However, if fine-tuning prioritizes retaining pre-trained knowledge, a smaller $\beta$ can be chosen to freeze more dimensions. ❸ **Trainable Component**: The remaining portion, with rank $r_1 = k_\alpha$, is designated as trainable. Unlike SVFit (Sun et al., 2024), which focuses solely on learning the distribution of singular values, we initialize the diagonal of the $S_p$ matrix with the top-$r_1$ singular values, while setting the remaining elements of the $r_1 \times r_1$ matrix to zero and making them trainable. This approach retains critical singular values while allowing for complex linear combinations between the left and right singular vectors, enhancing the model's expressive power.

By decomposing the matrix into three components, we enable fine-tuning within a much smaller matrix, significantly reducing the number of trainable parameters. This approach substantially lowers memory consumption and computational cost while preserving model performance.

### 3.3 THEORETICAL ANALYSIS

CERSA fine-tunes pre-trained weights with fewer parameters by decomposing them via SVD and freezing the $U$ and $V$ matrices for parameter efficiency. While methods like SVFit (Sun et al., 2024) and SVFT (Lingam et al., 2024) follow a similar approach, freezing $U$ and $V$ limits the model's expressiveness, making them suboptimal compared to full-parameter fine-tuning. CERSA overcomes this by introducing a trainable matrix $S_p$ initialized with singular values on the diagonal, reducing these constraints while maintaining memory efficiency. In the following, we establish the theoretical foundation of CERSA to show that its performance closely matches the full-parameter fine-tuning.

For any weight matrix $W \in \mathbb{R}^{m \times n}$ in a pre-trained model, we define its full-parameter fine-tuned counterpart in downstream tasks as $W'$. In the previous section, we removed the bottom 5%–10% of cumulative energy from $W$ by performing truncated SVD, as these correspond to insignificant

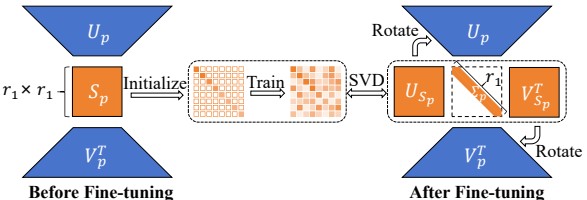 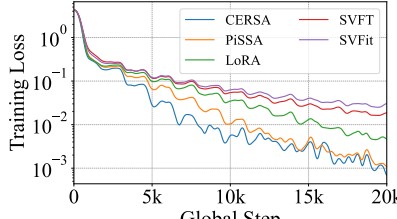

Figure 5: Training process of CERSA. The trainable core matrix $\boldsymbol{S}_p$ can be decomposed into $\boldsymbol{U}_{\boldsymbol{s}_p}$ and $\boldsymbol{V}_{\boldsymbol{s}_p}$, enabling fine-tuning by rotating the input and output bases without altering the subspace itself.

Figure 6: Loss curve of fine-tuning ViT-Large (Dosovitskiy, 2021) on CIFAR-100 (Krizhevsky & Hinton, 2009) using various methods.

features and noise in the neural network. This allows us to approximate the weight matrix as $\boldsymbol{W} \approx \boldsymbol{U}_p \boldsymbol{\Sigma}_p \boldsymbol{V}_p^T$. Similarly, the fine-tuned weight matrix can be approximated as $\boldsymbol{W}' \approx \boldsymbol{U}_p' \boldsymbol{\Sigma}_p' \boldsymbol{V}_p'^T$.

**Theorem 3.1.** *Given a matrix $\boldsymbol{M}$, applying the SVD, we have $\boldsymbol{M} = \boldsymbol{U} \boldsymbol{\Sigma} \boldsymbol{V}$. If there exists a pair of orthonormal bases $\boldsymbol{Q} = \{\boldsymbol{e}_1, \boldsymbol{e}_2, \ldots, \boldsymbol{e}_k\}$ and $\boldsymbol{Q}' = \{\boldsymbol{e}_1', \boldsymbol{e}_2', \ldots, \boldsymbol{e}_k'\}$ such that $Span(\boldsymbol{U}) = Span(\boldsymbol{Q})$, $Span(\boldsymbol{V}) = Span(\boldsymbol{Q}')$, there exists a matrix $\boldsymbol{S} \in \mathbb{R}^{k \times k}$ such that $\boldsymbol{M} = \boldsymbol{Q} \boldsymbol{S} \boldsymbol{Q}'^T$.*

Unlike SVFit (Sun et al., 2024) and SVFT (Lingam et al., 2024), which assume that $\boldsymbol{U}$ and $\boldsymbol{V}$ remain unchanged during fine-tuning, in practice, $\boldsymbol{U}$ and $\boldsymbol{V}$ are likely to be updated to adapt to downstream tasks. Therefore, we propose an alternative hypothesis: rather than $\boldsymbol{U}$ and $\boldsymbol{V}$ being strictly invariant, the truly preserved components are the principal subspaces of $\boldsymbol{U}_p'$ and $\boldsymbol{V}_p'$. This implies that the span of these sets of singular vectors remains unchanged, *i.e.*, $\text{Span}(\boldsymbol{U}_p') = \text{Span}(\boldsymbol{U}_p)$ and $\text{Span}(\boldsymbol{V}_p') = \text{Span}(\boldsymbol{V}_p)$. According to Theorem 3.1, since the principal subspaces remain the same before and after fine-tuning, there exists a transformation matrix $\boldsymbol{S}_p \in \mathbb{R}^{k \times k}$ such that $\boldsymbol{W}' \approx \boldsymbol{U}_p' \boldsymbol{\Sigma}_p' \boldsymbol{V}_p'^T = \boldsymbol{U}_p \boldsymbol{S}_p \boldsymbol{V}_p^T$. This suggests that rather than explicitly updating $\boldsymbol{U}_p$ and $\boldsymbol{V}_p$, we can freeze them and only update the intermediate matrix $\boldsymbol{S}_p$. This is mathematically equivalent to updating all three components: $\boldsymbol{U}_p$, $\boldsymbol{\Sigma}_p$, and $\boldsymbol{V}_p^T$, effectively removing the expressiveness constraints imposed by freezing $\boldsymbol{U}$ and $\boldsymbol{V}$. The proof of this theorem is provided in the Appendix (Sec. F.2).

To further illustrate the effect of a fully trainable matrix $\boldsymbol{S}_p \in \mathbb{R}^{k \times k}$, as shown in Fig. 5, we apply an additional SVD decomposition to the updated $\boldsymbol{S}_p$ after fine-tuning. The resulting $\boldsymbol{V}_{\boldsymbol{s}_p}^T$, an $r_1 \times r_1$ rotation matrix, rotates $\boldsymbol{V}_p^T$ to adjust the spatial distribution of input features while preserving the integrity of the subspace. Similarly, $\boldsymbol{U}_{\boldsymbol{s}_p}$ adjusts the rotation of $\boldsymbol{U}_p$, concentrating key features within the intermediate space along output directions essential for downstream tasks.

Additionally, our experiments (detailed in Appendix Sec. F.1) confirm that in most full-parameter fine-tuning downstream tasks, the principal subspaces of $\boldsymbol{W}$ and $\boldsymbol{W}'$ exhibit a Grassmann subspace similarity (Hu et al., 2022) of 99%~99.99%. This provides strong empirical evidence supporting the assumption that the principal subspaces of $\boldsymbol{W}$ remain nearly unchanged after fine-tuning.

Although this approach increases the number of trainable parameters from $k$ to $k^2$ (compared to SVFit (Sun et al., 2024)), the space saved through compression in the pre-trained model and the performance gains achieved make this increase a justifiable cost. The larger parameterization also enables stronger adaptability, as evidenced by a faster loss decrease during fine-tuning (see Fig. 6). At the same time, the benefits of fixing $\boldsymbol{U}_p$ and $\boldsymbol{V}_p$ are preserved, since the rotation matrix preserves the input and output subspace, only changing its basis representation. This ensures that features learned during pre-training remain intact, with only their distribution adjusted.

### 3.4 MEMORY EFFICIENCY

The primary goal of applying SVD to the pre-trained weight matrix is to reduce memory consumption during fine-tuning. Consequently, a compression rank threshold $b$ exists, below which memory savings are achieved only if $r < b$. Given a pre-trained weight matrix $\boldsymbol{W} \in \mathbb{R}^{m \times n}$, its truncated SVD decomposition produces: $\boldsymbol{U} \in \mathbb{R}^{m \times r}$, $\boldsymbol{\Sigma} \in \mathbb{R}^{r \times r}$, and $\boldsymbol{V} \in \mathbb{R}^{r \times n}$, such that $\boldsymbol{W} = \boldsymbol{U} \boldsymbol{\Sigma} \boldsymbol{V}$.

During training, memory usage consists of the frozen matrices $\boldsymbol{U}$ and $\boldsymbol{V}$, along with the trainable matrix $\boldsymbol{S}$, requiring $\mathcal{O}(mr + nr + r^2)$ storage. Additionally, since $\boldsymbol{S}$ is trainable, its gradient and

|  | FT | CERSA (Ours) | SVFit | SVFT | LoRA |
|---|---|---|---|---|---|
| Weights | $mn$ | $mr + nr + r^2$ | $2mn + m$ | $2mn + e$ | $mn + mr + nr$ |
| Gradients | $mn$ | $r^2$ | $r$ | $e$ | $mr + nr$ |
| Opt. states | $2mn$ | $2r^2$ | $2r$ | $2e$ | $2mr + 2nr$ |
| Total | $4mn$ | $mr + nr + 4r^2$ | $2mn + m + 3r$ | $2mn + 4e$ | $mn + 4mr + 4nr$ |

Table 1: Memory requirements. In SVFT, $e$ is the number of sparsified trainable parameters from the diagonal after SVD, where $e \ll mn$ but $e > m$.

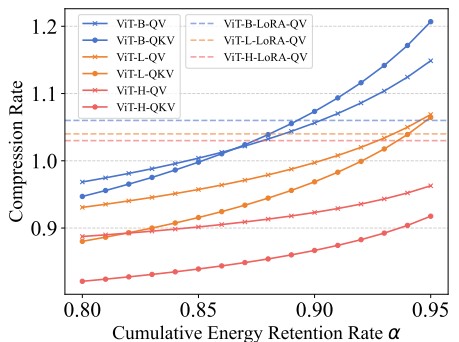

Figure 7: Comparison on ViT compression rates across various cumulative energy retention rates.

| Method | CIFAR-100 | RESISC45 | DTD | Average | Total Memory |
|---|---|---|---|---|---|
| CERSA(Q, V) | 94.0 | 95.8 | 82.1 | 90.6 | 1194.5 MB |
| CERSA(Q, K, V) | **94.4** | **96.1** | 82.5 | **91.0** | 1232.9 MB |
| CERSA(Q, K, V, P) | **94.5** | 96.0 | **82.6** | **91.0** | 1279.5 MB |
| CERSA(Q, K, V, P, UP, DN) | 93.8 | 94.9 | 81.6 | 90.1 | 1433.1 MB |

Table 2: Results of CERSA with various matrix type combinations for the ViT-Large model (Dosovitskiy, 2021). For the definitions of (Q, K, V, P, UP, DN), please refer to Fig. 3.

| Method | CIFAR-100 | RESISC45 | DTD | Average | Total Memory |
|---|---|---|---|---|---|
| Top-$r_1$ | **93.4** | **95.6** | **81.3** | **90.1** | 1112.7 MB |
| Bottom-$r_2$ | 92.6 | 94.7 | 79.9 | 89.1 | 1112.7 MB |

Table 3: Results of fine-tuning on Top-$r_1$ and Bottom-$r_2$ rank for the ViT-Large model (Dosovitskiy, 2021).

optimizer states contribute $\mathcal{O}(r^2)$ and $\mathcal{O}(2r^2)$, respectively, leading to a total memory requirement of $\mathcal{O}(mr + nr + 4r^2)$. Tab. 1 compares memory costs across different methods.

In contrast, the original weight matrix $\boldsymbol{W}$ requires $\mathcal{O}(mn)$ memory, excluding the additional $\mathcal{O}(3mn)$ for gradient and optimizer states, as we focus on reducing memory relative to the pre-trained parameters. The compression rate $c$ depends on the model's input-output dimensions, with a lower value indicating better memory efficiency: $c = \frac{mr + nr + 4r^2}{mn}$. Based on this, we compute the compression curves for three variants of the ViT model: Base, Large, and Huge, across cumulative energy retention rates ranging from 0.8 to 0.95, as shown in Fig. 7. The calculations are performed for two target module configurations: one with all three Q, K, and V matrices and another with only Q and V matrices. The dashed horizontal line in Fig. 7 represents the compression rate achieved by the LoRA method when fine-tuning only Q and V matrices with a rank of 32. The results indicate that for smaller models, such as ViT-Base, the compression rate is less favorable, possibly due to the limited embedding dimension. However, for larger models like ViT-Large and ViT-Huge (Dosovitskiy, 2021), there is considerably more room for compression, enabling greater memory efficiency.

## 4 EXPERIMENTS

We conduct extensive evaluations on both image classification and natural language understanding (NLU) tasks. Further experimental results, including subject-driven text-to-image generation and out-of-distribution evaluations, are provided in the appendix(see Sec. A.1 and Sec. E).

### 4.1 EXPERIMENTAL SETUP

**Baseline selection.** For baseline comparisons, we include full-parameter fine-tuning (FT), popular PEFT methods such as LoRA (Hu et al., 2022) and PiSSA (Meng et al., 2024), and weight-decomposition-based approaches like SVFit (Sun et al., 2024) and SVFT (Lingam et al., 2024).

**Model selection.** For image classification, we evaluate ViT-Base and ViT-Large (Dosovitskiy, 2021), pre-trained on ImageNet-21K (Deng et al., 2009). For the NLU experiment, we fine-tune DeBERTaV3-Base (He et al., 2023) to assess the fundamental capabilities of our method.

**Datasets.** For image classification, we assess our method on eight diverse datasets: CIFAR-100 (Krizhevsky & Hinton, 2009), EuroSAT (Helber et al., 2019), RESISC45 (Cheng et al., 2017), StanfordCars (Krause et al., 2013), FGVC Aircraft (Maji et al., 2013), DTD (Cimpoi et al., 2014),

| Method | CIFAR-100 | RESISC45 | DTD | Average | Total Memory |
|---|---|---|---|---|---|
| CERSA$_{\alpha=1,\beta=1}$ | 93.8 | **96.3** | 81.8 | 90.6 | 2519.6 MB |
| CERSA$_{\alpha=0.95,\beta=0.95}$ | **94.3** | 96.1 | **82.5** | **91.0** | 1232.9 MB |
| CERSA$_{\alpha=0.9,\beta=0.9}$ | 93.9 | 96.1 | 82.1 | 90.7 | 1122.5 MB |
| CERSA$_{\alpha=0.8,\beta=0.8}$ | 93.5 | 95.9 | 81.8 | 90.4 | 1020.6 MB |
| CERSA$_{\alpha=0.5,\beta=0.5}$ | 90.0 | 95.1 | 79.5 | 88.2 | 914.9 MB |
| CERSA$_{\alpha=0.95,\beta=0.9}$ | 94.0 | 96.0 | **82.5** | 90.8 | 1169.4 MB |
| CERSA$_{\alpha=0.95,\beta=0.8}$ | 93.8 | 96.1 | 82.2 | 90.7 | 1118.2 MB |
| CERSA$_{\alpha=0.95,\beta=0.5}$ | 92.9 | 95.2 | 80.3 | 89.5 | 1079.3 MB |

Table 4: Results of various cumulative energy retention rates for the ViT-Large model (Dosovitskiy, 2021) across the CIFAR-100 (Krizhevsky & Hinton, 2009), RESISC45 (Cheng et al., 2017), and DTD (Cimpoi et al., 2014) datasets.

| Method | CIFAR-100 | RESISC45 | DTD | Average | Total Memory |
|---|---|---|---|---|---|
| Layer-wise ($\alpha = \beta = 0.9$) | **93.9** | **96.1** | 82.1 | **90.7** | 1122.5 MB |
| Uniform ($r = 287$) | 93.7 | 95.6 | 81.4 | 90.2 | 1122.5 MB |

Table 5: Results of layer-wise and uniform rank for the ViT-Large model (Dosovitskiy, 2021).

| Method | CIFAR-100 | RESISC45 | DTD | Average | Total Memory |
|---|---|---|---|---|---|
| CERSA w/ Matrix | 93.9 | **96.1** | 82.1 | 90.7 | 1122.5 MB |
| CERSA w/ Array | 93.5 | 95.2 | 81.5 | 90.0 | 1045.5 MB |

Table 6: Results of CERSA with a trainable matrix or array for the ViT-Large model (Dosovitskiy, 2021).

CIFAR-10 (Krizhevsky & Hinton, 2009), and OxfordPets (Parkhi et al., 2012). These datasets span a variety of classification tasks, including general object classification, fine-grained classification, remote sensing image classification, and texture classification.

For NLU, we evaluate our method on eight datasets from the GLUE benchmark (Wang et al., 2019): MNLI, MRPC, RTE, CoLA, SST-2, QNLI, QQP, and STS-B. These datasets cover a broad spectrum of NLU tasks, including textual entailment, paraphrase detection, sentiment analysis, question-answer matching, and semantic textual similarity.

**Metrics.** For image classification, we report accuracy across all datasets. For NLU, we report overall matched and mismatched accuracy on MNLI, Matthew's correlation on CoLA, Pearson correlation on STS-B, and accuracy on the remaining datasets. Higher values indicate better performance.

## 4.2 ABLATION STUDY

**Impact of the matrix type.** We study the trade-off between performance and memory when fine-tuning different matrix types. As shown in Tab. 2, adapting Q, K, and V achieves the best balance of accuracy and efficiency. In contrast, adding P, UP, or DN increases memory cost and even reduces performance. This is mainly because: (i) these matrices require higher ranks to preserve cumulative energy (Fig. 3), making them less memory-efficient; and (ii) modifying them disrupts pre-trained feature representations, leading to overfitting or weaker generalization. Thus, restricting fine-tuning to Q, K, and V provides the optimal trade-off.

**Impact of top-$r_1$ versus bottom-$r_2$ ranks.** To assess the effect of fine-tuning major versus residual components, we compare $S_p$ trained on the top-$r_1$ and bottom-$r_2$ components (with $r_1 = r_2$, $\alpha = 0.95$). As shown in Tab. 3, adapting the top components consistently yields higher accuracy, validating our design. Moreover, the OOD results in the appendix(see Sec. E) show that CERSA surpasses both LoRA and FT, confirming that it preserves rather than distorts the original knowledge.

**Impact of the cumulative energy retention rate.** We investigate how different retention rates affect fine-tuning by training ViT-Large (Dosovitskiy, 2021) under various $\alpha$ and $\beta$ configurations (Tab. 4). The first five settings use $\alpha = \beta$ decreasing from 1 to 0.5, while the last three fix $\alpha = 0.95$ and vary $\beta$ to adjust the trainable subspace size. Among them, CERSA$_{\alpha=0.95,\beta=0.95}$ achieves the best overall accuracy. Performance remains stable even at $\alpha = \beta \approx 0.8$ (90% of pre-trained memory), but drops sharply when reduced to 0.5 or lower.

**Impact of layer-wise versus uniform.** We compare layer-wise CERSA, which selects singular values by each layer's cumulative energy retention rate, with uniform CERSA, which fixes the rank at 287 for all layers. As shown in Tab. 5, despite identical memory consumption, layer-wise CERSA consistently outperforms the uniform variant, demonstrating the effectiveness of exploiting layer-specific retention rates.

**Impact of tuning a matrix versus array.** We analyze the impact of defining the $S$ matrix as either a matrix or an array (as in SVFit (Sun et al., 2024)) on fine-tuning performance under the same CERSA configuration $\alpha = 0.9, \beta = 0.9$ for Q, K, and V. As shown in Tab. 6, although arrays significantly reduce memory usage, they result in a substantial drop in performance, highlighting that matrix-based fine-tuning has much better expressiveness.

| Method | Memory | CIFAR-100 | EuroSAT | RESISC45 | StanfordCars | FGVC-Aircraft | DTD | CIFAR-10 | OxfordPets | Average |
|--------|--------|-----------|---------|----------|--------------|---------------|-----|----------|------------|---------|
| FT | 4629.8 MB | 93.6 | 99.0 | 96.4 | 88.9 | 68.3 | 81.8 | 99.2 | 94.4 | 90.2 |
| LoRA | 1229.9 MB | **94.9** | 99.0 | 94.7 | 80.3 | 54.5 | 81.5 | 99.1 | 94.8 | 87.4 |
| PiSSA | 1229.9 MB | 93.6 | 98.6 | 95.7 | 86.7 | 62.6 | 81.8 | 98.8 | **95.7** | 89.2 |
| SVFit | 1351.5 MB | 93.9 | 98.7 | 95.2 | 83.3 | 57.8 | 81.5 | **99.3** | 93.4 | 88.7 |
| SVFT | 1355.8 MB | 93.7 | 98.8 | 95.4 | 84.9 | 63.7 | 82.3 | **99.3** | 93.3 | 88.9 |
| CERSA | 1232.3 MB | 94.3 | **99.1** | **96.1** | 87.6 | **71.1** | **82.5** | **99.3** | 94.9 | **90.6** |

Table 7: Comparison of various fine-tuning methods on eight image classification datasets using ViT-Large (Dosovitskiy, 2021). Methods include LoRA (Hu et al., 2022), PiSSA (Meng et al., 2024), SVFit (Sun et al., 2024), and SVFT (Lingam et al., 2024). Bold scores indicate the highest accuracy among PEFT methods, while underlined scores indicate that full-parameter fine-tuning (FT) achieves the best performance.

| Method | Memory | MNLI | MRPC | STS-B | RTE | SST-2 | QNLI | QQP | CoLA | Average |
|--------|--------|------|------|-------|-----|-------|------|-----|------|---------|
| FT | 2814.0 MB | 89.9 | 89.5 | 91.6 | 83.8 | 95.6 | 94.0 | 92.4 | 69.2 | 88.3 |
| LoRA | 730.5 MB | **90.7** | 90.0 | 91.6 | 85.2 | 95.0 | 93.9 | 92.0 | 69.8 | 88.5 |
| PiSSA | 730.5 MB | 90.4 | 91.7 | **91.9** | 87.0 | 95.9 | 94.3 | 92.3 | **72.6** | **89.5** |
| SVFit | 1096.3 MB | 89.7 | 88.8 | 91.8 | **87.4** | 95.4 | 94.3 | 90.2 | 71.0 | 88.6 |
| SVFT | 1108.6 MB | 90.0 | 89.0 | 91.8 | 87.2 | 95.4 | 94.3 | 91.5 | **72.6** | 89.0 |
| CERSA | 728.6 MB | 90.3 | **92.0** | 91.7 | 87.0 | **96.0** | **94.4** | **92.4** | 72.3 | **89.5** |

Table 8: Comparison of different methods on the GLUE benchmark using the DeBERTaV3-Base model (He et al., 2023). Methods include LoRA (Hu et al., 2022), PiSSA (Meng et al., 2024), SVFit (Sun et al., 2024), and SVFT (Lingam et al., 2024).

## 4.3 COMPARISON ON IMAGE CLASSIFICATION TASKS

Experimental results for the ViT-Large model (Dosovitskiy, 2021) are presented in Tab. 7. With ViT-Large (Dosovitskiy, 2021), CERSA achieves an average accuracy of 90.6%, outperforming full-parameter fine-tuning (90.2%) and significantly surpassing other PEFT methods like SVFT (Lingam et al., 2024) (88.9%) and SVFit (Sun et al., 2024) (88.7%). Notably, CERSA excels on fine-grained classification tasks, particularly for datasets like StanfordCars (Krause et al., 2013) and FGVC Aircraft (Maji et al., 2013), highlighting its capability to capture intricate details. Besides, CERSA matches or exceeds full-parameter fine-tuning on general datasets like CIFAR-100 (Krizhevsky & Hinton, 2009), EuroSAT (Helber et al., 2019), and RESISC45 (Cheng et al., 2017), demonstrating its strong generalization and adaptability across diverse tasks.

## 4.4 COMPARISON ON NLU TASKS

Tab. 8 compares fine-tuning strategies on eight GLUE datasets using DeBERTaV3-Base (He et al., 2023). CERSA achieves the highest average score (89.5%), outperforming both full-parameter fine-tuning and other PEFT methods, and sets new state-of-the-art results on multiple datasets, including MRPC, SST-2, QNLI, and QQP. For the remaining tasks, it also attains competitive performance compared with state-of-the-art methods PiSSA (Meng et al., 2024), SVFit (Sun et al., 2024), and SVFT (Lingam et al., 2024). This highlights the effectiveness of CERSA in leveraging pre-trained representations with minimal computational overhead, making it a strong choice for NLU tasks.

## 5 CONCLUSION

We propose CERSA, a memory- and parameter-efficient fine-tuning method that performs layer-wise rank selection based on the cumulative energy retention of pre-trained weights, enabling adaptation within the principal subspace. We prove that CERSA achieves performance comparable to full fine-tuning, and extensive experiments on image classification and language understanding show it outperforms or matches state-of-the-art PEFT methods while reducing memory.

**Limitation and Future Work.** Since CERSA constrains fine-tuning within the principal subspace of pre-trained weights, its performance may degrade when the downstream task significantly deviates from the knowledge captured during pre-training. In the future, we plan to extend CERSA's capabilities by dynamically adjusting its learned subspace during fine-tuning, thereby enhancing its adaptability and performance across a broader range of downstream tasks.

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
