APPENDIX

# A ADDITIONAL EXPERIMENTAL RESULTS

## A.1 SUBJECT-DRIVEN TEXT-TO-IMAGE GENERATION

For subject-driven text-to-image generation, we fine-tune models using selected samples from the DreamBooth (Ruiz et al., 2023) and CustomConcept101 (Kumari et al., 2022) datasets. Each subject sample contains 5 to 6 images captured from different angles and contexts. We compare full-parameter fine-tuning (FT), LoRA (Hu et al., 2022), and our proposed CERSA on this task.

As shown in Fig. 1, we evaluate subject-driven generation across multiple domains, including scene composition, material modification, and artistic style transfer:

- **Scene composition.** When placing a sports car in front of the Eiffel Tower or on a New York street, CERSA captures both background details and subject fidelity more accurately than LoRA, producing results that more closely resemble FT.

- **Material modification.** Applying glass and silver textures to a duck toy highlights CERSA's strength: it preserves the subject's original shape and features while achieving consistent material transfer. In contrast, LoRA and FT often distort shapes or fail to maintain color/material consistency.

- **Style transfer.** When adapting a dog's image into the styles of Vincent van Gogh and Leonardo da Vinci, all three methods demonstrate recognizable style transfer, but CERSA produces visuals that align more closely with FT while avoiding artifacts.

**Quantitative comparison.** We use CLIPScore (Hessel et al., 2022) to assess prompt-image alignment. CERSA achieves the highest average CLIPScore(**32.75**), outperforming LoRA (31.88) and FT (32.35), indicating better generation quality.

Overall, these results demonstrate that CERSA achieves high-quality subject-driven image generation, consistently surpassing LoRA and closely matching or even exceeding the performance of full-parameter fine-tuning, while being significantly more memory- and parameter-efficient.

## A.2 IMAGE CLASSIFICATION

In addition to testing the performance of CERSA on ViT-Large (Dosovitskiy, 2021), we also test it on ViT-Base (Dosovitskiy, 2021). With ViT-Base Tab. 1, CERSA achieve an average accuracy of 89.0% across eight datasets, outperforming full parameter fine-tuning (**FT**) (86.5%) and significantly surpassing other PEFT methods like LoRA (Hu et al., 2022) (77.6%), PiSSA (Meng et al., 2024) (84.2%), SVFT (Lingam et al., 2024) (84.6%), and SVFit (Sun et al., 2024) (83.7%). It also excels on fine-grained classification tasks, particularly Stanford Cars and FGVC Aircraft (Maji et al., 2013), and matches or exceeds FT on general datasets like CIFAR-10 (Krizhevsky & Hinton, 2009), Oxford Pets (Parkhi et al., 2012), and DTD (Cimpoi et al., 2014), demonstrating strong generalization. Compared to ViT-Large (Dosovitskiy, 2021), the performance advantage of our method is more obvious on ViT-base (Dosovitskiy, 2021), but the compression rate is not as good as that of the large model.

Additionally, in Tab. 2, we evaluate the performance of all image classification tasks on ViT-Base (Dosovitskiy, 2021) under different settings of the cumulative energy retention ratio. In the first set of experiments, we set $\alpha = \beta$, which means that the entire principal subspace corresponding to the cumulative energy is fine-tuned. We test performance under different cumulative energy retention ratios $\{0.95, 0.9, 0.8, 0.5\}$. In the second set of experiments, we fix $\alpha = 0.95$ and examine the performance of fine-tuning only a portion of the principal subspace, with $\beta$ set to 0.9, 0.8, and 0.5, respectively.

In the first set of experiments, we observe that the average performance drop is minimal (only 1.4%) when the cumulative energy retention ratio ranges from 0.95 to 0.8. Only when the ratio decreased to 0.5 did a significant performance decline occur. This indicates that we have ample room to trade off a slight performance loss for a substantial reduction in overall memory consumption. In the second set

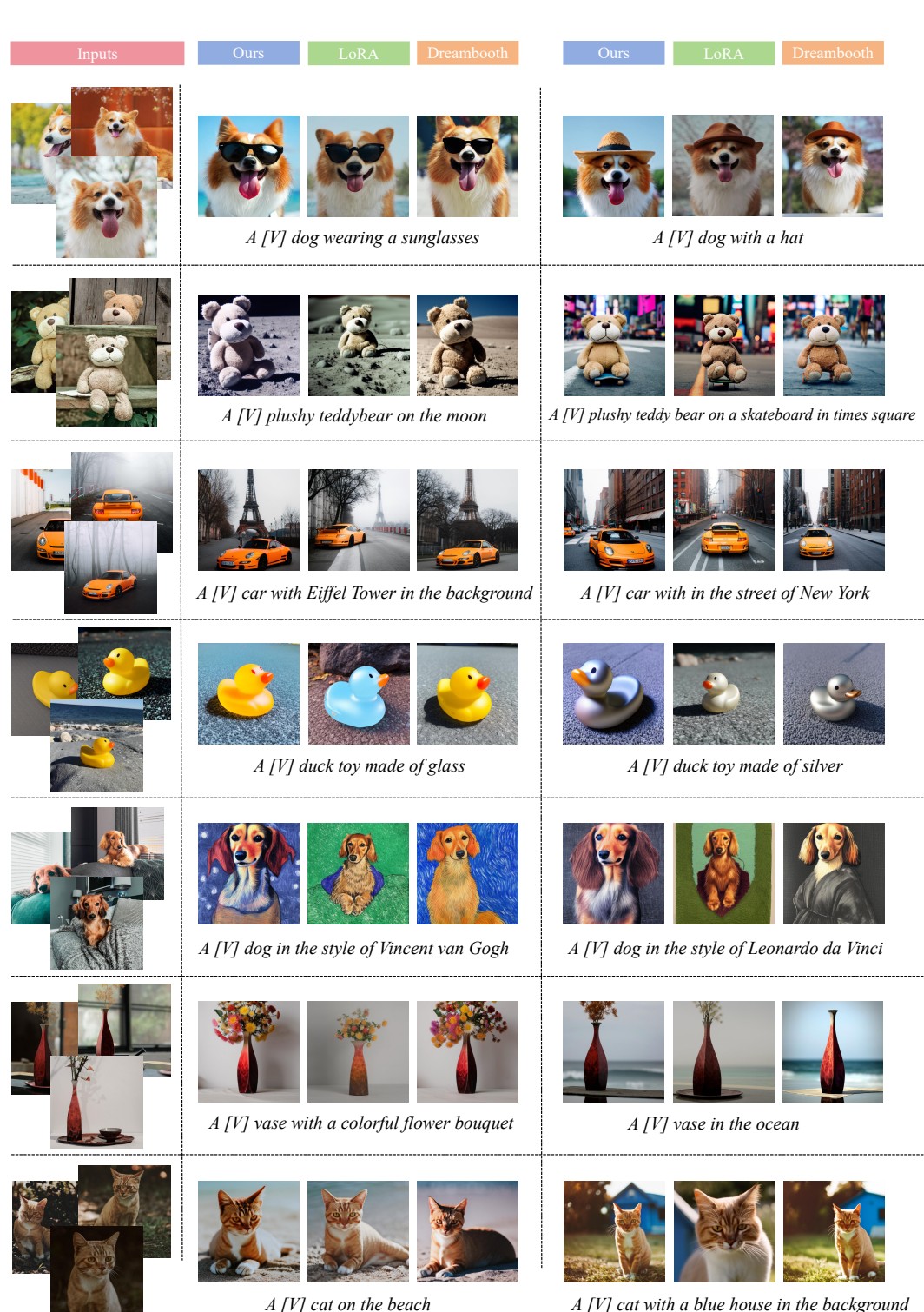

Figure 1: Results of visual comparison generated by the subject-driven fine-tuned diffusion model using the proposed CERSA, LoRA (Hu et al., 2022), and DreamBooth (Ruiz et al., 2023).

| Method | CIFAR-100 | EuroSAT | RESISC45 | StanfordCars | FGVC-Aircraft | DTD | CIFAR-10 | OxfordPets | Average |
|---|---|---|---|---|---|---|---|---|---|
| FT | 92.4 | 99.1 | 96.1 | 79.8 | 54.8 | 77.7 | 98.9 | 93.1 | 86.5 |
| LoRA | 92.0 | 98.4 | 92.7 | 45.5 | 25.2 | 75.0 | **98.8** | 93.1 | 77.6 |
| PiSSA | 91.2 | 98.7 | 95.5 | 67.1 | 47.6 | 78.7 | 98.6 | **95.9** | 84.2 |
| SVFit | 91.6 | 98.6 | 93.0 | 67.2 | 47.9 | 80.5 | **98.8** | 92.3 | 83.7 |
| SVFT | 91.2 | 98.5 | 92.4 | 67.5 | 56.2 | 79.8 | 98.7 | 92.5 | 84.6 |
| CERSA | **92.1** | **98.9** | **95.6** | **83.9** | **68.2** | **81.2** | 98.8 | 93.2 | **89.0** |

Table 1: Comparison of various fine-tuning methods on eight image classification datasets using ViT-Base (Dosovitskiy, 2021). Methods include LoRA (Hu et al., 2022), PiSSA (Meng et al., 2024), SVFit (Sun et al., 2024), and SVFT (Lingam et al., 2024). Bold scores indicate the highest accuracy among PEFT methods, while underlined scores indicate that full-parameter fine-tuning (FT) achieves the best performance.

| | CIFAR-100 | EuroSAT | RESISC45 | StanfordCars | FGVC-Aircraft | DTD | CIFAR-10 | OxfordPets | Average |
|---|---|---|---|---|---|---|---|---|---|
| $\text{CERSA}_{\alpha=1,\beta=1}$ | 91.3 | 97.6 | 85.5 | 72.8 | 64.6 | 78.9 | 98.6 | 92.3 | 85.2 |
| $\text{CERSA}_{\alpha=0.95,\beta=0.95}$ | **92.1** | **98.9** | **95.6** | **83.9** | **68.2** | **81.2** | 98.8 | **93.2** | 89.0 |
| $\text{CERSA}_{\alpha=0.9,\beta=0.9}$ | **92.1** | 98.6 | 95.3 | 83.5 | 68.2 | 80.3 | 98.6 | **93.2** | 88.7 |
| $\text{CERSA}_{\alpha=0.8,\beta=0.8}$ | 91.1 | 98.1 | 94.9 | 80.2 | 67.4 | 78.1 | 98.5 | 92.6 | 87.6 |
| $\text{CERSA}_{\alpha=0.5,\beta=0.5}$ | 83.3 | 95.4 | 91.7 | 62.9 | 50.7 | 67.4 | 96.3 | 87.7 | 79.4 |
| $\text{CERSA}_{\alpha=0.95,\beta=0.95}$ | 92.1 | **98.9** | **95.6** | **83.9** | 68.2 | **81.2** | 98.8 | **93.2** | 89.0 |
| $\text{CERSA}_{\alpha=0.95,\beta=0.9}$ | 92.1 | 98.6 | 95.3 | 83.7 | 69.8 | 80.5 | 98.7 | 93.1 | 88.9 |
| $\text{CERSA}_{\alpha=0.95,\beta=0.8}$ | **92.3** | 98.1 | 95.1 | 81.5 | **70.4** | 79.0 | 98.6 | **93.2** | 88.5 |
| $\text{CERSA}_{\alpha=0.95,\beta=0.5}$ | 91.5 | 95.4 | 94.3 | 75.8 | 60.2 | 77.6 | 98.4 | **93.2** | 85.8 |

Table 2: Evaluation results of CERSA on eight image classification datasets under different $\alpha$ and $\beta$ settings using ViT-Base (Dosovitskiy, 2021).

| Dataset | CIFAR-100 | EuroSAT | RESISC45 | StanfordCars | FGVC-Aircraft | DTD | CIFAR-10 | OxfordPets |
|---|---|---|---|---|---|---|---|---|
| Attention Dropout | 0.1 | 0.1 | 0.1 | 0 | 0.1 | 0.1 | 0.1 | 0 |
| Weight Decay | 1e-3 | 1e-3 | 1e-3 | 0.01 | 1e-3 | 1e-3 | 1e-3 | 0.01 |
| LR | 1e-4 | 8e-5 | 1e-3 | 1e-3 | 2e-3 | 3e-4 | 1e-4 | 1e-4 |
| LR (Classifier) | 1e-3 | 5e-4 | 3e-3 | 3e-3 | 6e-3 | 1e-3 | 1e-3 | 1e-3 |

Table 3: Hyperparameter settings for ViT-Large (Dosovitskiy, 2021) across different datasets for image classification experiments. LR: Learning Rate.

of experiments, we found that reducing $\beta$ from 0.95 to 0.8 results in only a performance drop of 0.5%, and even at $\beta = 0.5$, the performance decrease is limited to 3.2%. This suggests that fine-tuning only the most principal part of the preserved subspace allows for a more parameter-efficient approach while incurring only a minor performance loss.

# B  MORE IMPLEMENTATION DETAILS

**Experimental Environment.**    All experiments were conducted on an NVIDIA L40 GPU using the PyTorch framework (Paszke et al., 2019) and Hugging Face's `Transformers` library (Wolf et al., 2020) for fine-tuning.

**Settings for Image Classification.**    For image classification, we fine-tune ViT-Base and ViT-Large (Dosovitskiy, 2021) on the Query (**Q**), Key (**K**), and Value (**V**) matrices within the attention module. In our method, CERSA, we set a cumulative energy retention rate of $\alpha = \beta = 0.95$ across all fine-tuning tasks. For comparison, we configure LoRA (Hu et al., 2022) and PiSSA (Meng et al., 2024) with a rank of 32, a commonly chosen value that balances performance and the number of trainable parameters.

For SVFit (Sun et al., 2024), we adhere to the recommended configuration of the original paper, using a rank of 768 for all models. Similarly, for SVFT (Lingam et al., 2024), we adopt the best-performing settings. We use the AdamW optimizer (Loshchilov & Hutter, 2017) with a fixed batch size of 32 and a linear scheduler incorporating a warm-up ratio of 0.08. For further details on hyperparameter settings, see Tab. 3 and Tab. 4.

**Settings for Text-to-Image Generation.**    For the subject-driven text-to-image generation task, We use Stable Diffusion v2-1-base (Rombach et al., 2022) as the pre-trained model and apply DreamBooth (Ruiz et al., 2023) for subject-driven text-to-image fine-tuning. We follow the setup of DreamBooth (Ruiz et al., 2023) to evaluate CERSA's fine-tuning. This ensures that the method

| Dataset | CIFAR-100 | EuroSAT | RESISC45 | StanfordCars | FGVC-Aircraft | DTD | CIFAR-10 | OxfordPets |
|---|---|---|---|---|---|---|---|---|
| Attention Dropout | 0.1 | 0.1 | 0.1 | 0 | 0.1 | 0.1 | 0.1 | 0 |
| Weight Decay | 1e-3 | 1e-3 | 1e-3 | 0.01 | 1e-3 | 1e-3 | 1e-3 | 0.01 |
| LR | 2e-4 | 1e-4 | 2e-3 | 2e-3 | 1e-3 | 2e-4 | 2e-4 | 2e-4 |
| LR (Classifier) | 1e-3 | 5e-4 | 5e-3 | 5e-3 | 5e-3 | 1e-3 | 1e-3 | 1e-3 |

Table 4: Hyperparameter settings for ViT-Base (Dosovitskiy, 2021) across different datasets for image classification experiments. LR: Learning Rate.

| Dataset | MNLI | SST-2 | MRPC | CoLA | QNLI | QQP | RTE | STS-B |
|---|---|---|---|---|---|---|---|---|
| Max Seq. Len. | 256 | 128 | 320 | 64 | 512 | 320 | 320 | 128 |
| Epochs | 8 | 16 | 30 | 15 | 10 | 8 | 20 | 15 |
| Batch Size | 16 | 32 | 16 | 16 | 32 | 16 | 16 | 32 |
| Classifier Dropout | 0.15 | 0 | 0 | 0.1 | 0.1 | 0.2 | 0.2 | 0.2 |
| Weight Decay | 0 | 0.01 | 0.01 | 0 | 0.01 | 0.01 | 0.01 | 0.1 |
| LR | 1e-4 | 1e-4 | 2e-4 | 1e-4 | 1e-4 | 1e-4 | 2e-4 | 2e-4 |
| LR(Classifier) | 3e-4 | 3e-4 | 4e-4 | 3e-4 | 3e-4 | 3e-4 | 4e-4 | 4e-4 |

Table 5: Hyperparameter settings for DeBERTa-V3-Base (He et al., 2023) across different datasets for NLU experiments. LR: Learning Rate.

| Methods | Trainable Parameter (M) | Trainable Ratio (%) | Weights Memory (MB) | Optimizer State Memory (MB) | Gradient Memory (MB) | Total Memory (MB) |
|---|---|---|---|---|---|---|
| FT | 303.3 | 100 | **1157.7** | 2314.4 | 1157.7 | 4629.8 |
| LoRA (Hu et al., 2022)($r = 8$) | 0.8 | 0.3 | 1161.9 | 9.4 | 4.7 | 1175.9 |
| LoRA (Hu et al., 2022)($r = 32$) | 3.2 | 1.0 | 1175.4 | 36.4 | 18.2 | 1229.9 |
| SVFit (Sun et al., 2024) | 0.04 | 0.02 | 1349.8 | 1.1 | 0.6 | 1351.5 |
| SVFT (Lingam et al., 2024) | 0.12 | 0.06 | 1350.9 | 3.3 | 1.7 | 1355.8 |
| CERSA$_{\alpha=\beta=0.95}$ | 10.5 | 3.6 | 1111.4 | 80.5 | 40.3 | 1232.2 |
| CERSA$_{\alpha=\beta=0.92}$ | 8.1 | 2.8 | 1069.0 | 59.0 | 29.5 | **1157.5** |
| CERSA$_{\alpha=\beta=0.9}$ | 6.3 | 2.3 | 1048.6 | 49.3 | 24.6 | **1122.5** |
| CERSA$_{\alpha=\beta=0.85}$ | 4.3 | 1.6 | 1011.4 | 33.3 | 16.7 | **1061.4** |
| CERSA$_{\alpha=\beta=0.8}$ | 3.0 | 1.2 | 985.2 | 23.6 | 11.8 | **1020.6** |

Table 6: Memory consumption comparison across various methods with different settings.

captured subject-specific details while preserving pre-trained knowledge. We compare CERSA with full-parameter DreamBooth (Ruiz et al., 2023) and LoRA (Hu et al., 2022), evaluating image quality and textual alignment.

In our implementation, we replace all linear layers in the UNet (Ronneberger et al., 2015) and the attention modules of the CLIP (Radford et al., 2021) text encoder with CERSA. The cumulative energy retention rate is set to $\alpha = \beta = 0.95$. For LoRA (Hu et al., 2022), we insert the adapters into the same layers with a rank of 32. For full-parameter fine-tuning, we made all these layers trainable. The VAE(variational autoencoder) module remains frozen in all methods. We use the AdamW (Loshchilov & Hutter, 2017) optimizer and a constant scheduler. To ensure fairness in the inference stage, we use identical random seeds, inference steps, and guidance scales across all methods, preventing variations due to different parameter settings.

**Settings for NLU Experiments.** For the NLU experiments, we fine-tune the Q, K, and V matrices in DeBERTa-v3-base (He et al., 2023). The adapter rank for LoRA (Hu et al., 2022) and PiSSA (Meng et al., 2024) is set to 32. SVFit (Sun et al., 2024) and SVFT (Lingam et al., 2024) use the same settings as in the image classification experiments. To ensure fairness, we follow SVFT's (Lingam et al., 2024) max sequence length settings. We use the AdamW (Loshchilov & Hutter, 2017) optimizer and employ a linear scheduler with a warm-up ratio of 0.08. For detailed hyperparameters, see Tab. 5.

## C  PERFORMANCE ON MEMORY CONSUMPTION

Tab. 6 compares the parameter and memory efficiency of various fine-tuning methods. We exclude activation and dataset-related memory usage, as they remain largely independent of the fine-tuning approach. Thus, total memory refers to the sum of the weight size, the gradient size, and the size of the optimizer parameter. Besides, in Tab. 6, we report the number of trainable parameters in millions (M).

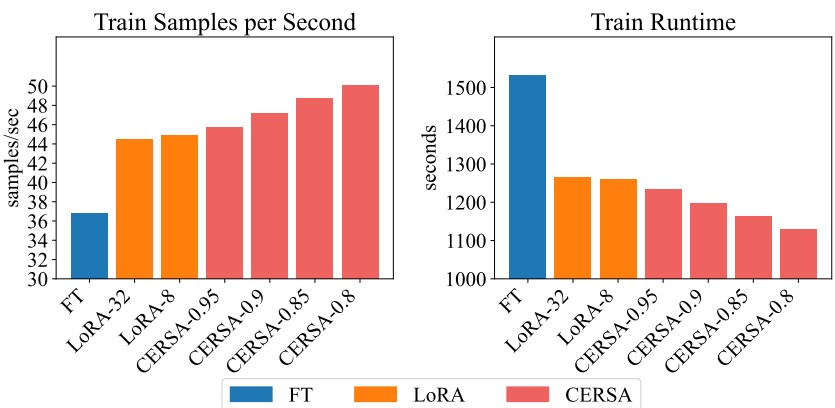

Figure 2: Training throughput and training time of fine-tuning ViT-Large (Dosovitskiy, 2021) on the DTD (Cimpoi et al., 2014) dataset under various configurations.

Full-parameter fine-tuning (FT) updates all model parameters (303.3 M trainable parameters), resulting in a substantial total memory consumption of 4629.8 MB. This high memory demand makes FT impractical for resource-constrained environments.

LoRA (Hu et al., 2022), with ranks of 8 and 32, significantly reduces the number of trainable parameters to 0.8 M and 3.2 M, respectively. However, its total memory consumption remains considerable – 1175.9 MB for rank=8 and 1229.9 MB for rank=32 – exceeding the memory footprint of the pre-trained weights due to additional optimizer state and gradient storage. Similarly, SVFit (Sun et al., 2024) achieves high parameter efficiency with only 0.04 M of trainable parameters yet still requires 1351.5 MB of total memory, primarily due to the storage overhead of full singular vector matrices.

The proposed CERSA method provides a flexible solution for memory and parameter-efficient fine-tuning by adjusting the cumulative energy retention rate, enabling different levels of efficiency based on memory constraints. For example, with a relatively ample memory budget, setting the retention rate to $\alpha = \beta = 0.95$ yields better performance. At $\alpha = \beta = 0.92$, CERSA maintains a memory footprint equivalent to the pre-trained weights during fine-tuning. When reduced to $\alpha = \beta = 0.8$, it retains 3.0 M trainable parameters comparable to LoRA (Hu et al., 2022) (rank=32) while significantly lowering total memory consumption to 1020.6 MB (while LoRA (Hu et al., 2022) uses up to 1229.9 MB).

Although not as parameter-efficient as SVFit (Sun et al., 2024) and SVFT (Lingam et al., 2024), CERSA excels in overall memory efficiency, even with more trainable parameters. This makes it particularly advantageous for fine-tuning large-scale models in memory-constrained environments. Additionally, its adjustable cumulative energy retention rate allows for customized trade-offs, making CERSA a versatile solution that outperforms other PEFT methods in total memory consumption while maintaining competitive performance.

# D PERFORMANCE ON SPEED

CERSA decomposes the pre-trained weight matrix into three components: $U_p$, $S_p$, and $V_p$. For simplicity, we assume that CERSA is configured with $\alpha = \beta = 0.95$. Compared to the original weight matrix $W$, this decomposition introduces more granular matrix computations. However, since the size of the matrices involved in computation is significantly reduced, the overall computational cost is also reduced. To evaluate the actual impact on fine-tuning, we design experiments to measure *training throughput* and *training time*.

To eliminate the impact of dataset pre-processing and batch size on computation time and throughput, we fix the batch size at 32 and the number of epochs at 15. Fine-tuning is performed on ViT-Large (Dosovitskiy, 2021) across full-parameter fine-tuning, LoRA (Hu et al., 2022), and CERSA.

Experimental results show that our method achieves a comparable or superior training efficiency to LoRA while significantly outperforming FT in terms of speed. As shown in Fig. 2, LoRA (*r*=32,

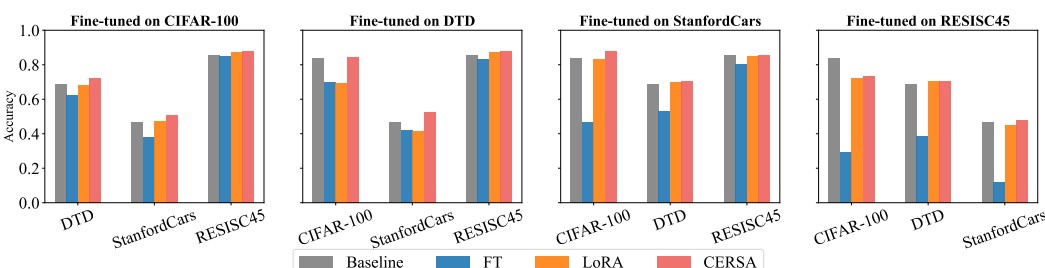

Figure 3: Out-of-distribution evaluation on various tasks.

| Method | FT | LoRA (Hu et al., 2022) | CERSA |
|---|---|---|---|
| Average Forgetting Rate | 17.8% | 2.3% | -1.5% |

Table 7: Average forgetting rate of FT, LoRA (Hu et al., 2022) and CERSA on the four datasets (CIFAR-100 (Krizhevsky & Hinton, 2009), DTD (Cimpoi et al., 2014), StanfordCars (Krause et al., 2013), and RESISC45 (Cheng et al., 2017))

$r$=8) improves throughput by about 30% over FT. CERSA, across all cumulative energy retention rates {0.95, 0.9, 0.85, 0.8}, slightly exceeds LoRA's efficiency, demonstrating that the cumulative energy retention decomposition of the weight matrix effectively reduces computational complexity while preserving model capacity. Despite introducing more granular matrix multiplications, the significantly reduced dimensionality effectively lowers the computational cost. As a result, CERSA matches or even surpasses LoRA in fine-tuning speed.

## E  PERFORMANCE ON OUT-OF-DISTRIBUTION TASKS

During full-parameter fine-tuning, the model gradually forgets core features from the pre-training data as its parameter space shifts significantly. In contrast, CERSA restricts updates to $S_p$, adjusting only the most critical feature subspace while ensuring that the principal subspace remains unaffected by less important dimensions. This preserves essential pre-trained knowledge.

Out-of-distribution(OOD) performance is a crucial indicator of knowledge retention, as previously studied in (Hendrycks & Gimpel, 2017) and (Kumar et al., 2022). Fig. 3 shows the OOD performance of models fine-tuned on each of the four datasets (CIFAR-100 (Krizhevsky & Hinton, 2009), DTD (Cimpoi et al., 2014), StanfordCars (Krause et al., 2013), and RESISC45 (Cheng et al., 2017)), with accuracy evaluated on the remaining three datasets. We compare FT, LoRA (Hu et al., 2022), and our proposed CERSA method. The gray bars indicate the model's original performance before fine-tuning, serving as a reference for relative performance degradation.

Across all fine-tuning settings, CERSA consistently achieves superior OOD performance compared to FT and LoRA (Hu et al., 2022). Specifically, when fine-tuned on CIFAR-100 (Krizhevsky & Hinton, 2009) (leftmost subplot), CERSA maintains a higher average OOD accuracy than LoRA (Hu et al., 2022) and FT, suggesting that it better preserves pre-trained knowledge for handling novel tasks such as DTD (Cimpoi et al., 2014) and StanfordCars (Krause et al., 2013), or even leverages knowledge from CIFAR-100 (Krizhevsky & Hinton, 2009). A similar trend is observed in the DTD (Cimpoi et al., 2014) fine-tuning scenario (second subplot), where CERSA demonstrates stronger retention of pre-trained features, particularly on CIFAR-100 (Krizhevsky & Hinton, 2009).

In our experiments, fine-tuning is performed on one dataset while accuracy is evaluated on the remaining three. The average forgetting rate is defined as the ratio of the average accuracy drop in the three out-of-distribution tasks compared to the baseline accuracy of the pre-trained model after fine-tuning on a specific task. As shown in Tab. 7, these results highlight CERSA's ability to mitigate catastrophic forgetting by retaining key representations learned during pre-training, thereby preserving higher accuracy on tasks not directly involved in fine-tuning.

| Dataset | CIFAR-100 | EuroSAT | RESISC45 | StanfordCars |
|---|---|---|---|---|
| Q | 99.69%/99.65% | 99.94%/99.94% | 99.81%/99.79% | 99.95%/99.94% |
| K | 99.76%/99.74% | 99.96%/99.96% | 99.86%/99.85% | 99.94%/99.94% |
| V | 99.58%/99.58% | 99.91%/99.91% | 99.79%/99.79% | 99.92%/99.92% |
| Dataset | FGVC-Aircraft | DTD | CIFAR-10 | OxfordPets |
| Q | 99.76%/99.73% | 99.91%/99.90% | 99.68%/99.63% | 99.96%/99.94% |
| K | 99.78%/99.76% | 99.94%/99.94% | 99.74%/99.72% | 99.94%/99.93% |
| V | 99.69%/99.70% | 99.85%/99.86% | 99.55%/99.55% | 99.89%/99.90% |

Table 8: Principal subspace similarity between the Q, K, and V matrices of ViT-Large (Dosovitskiy, 2021) pre-trained on ImageNet-21K (Deng et al., 2009) and the fine-tuned weights on various downstream image classification tasks.

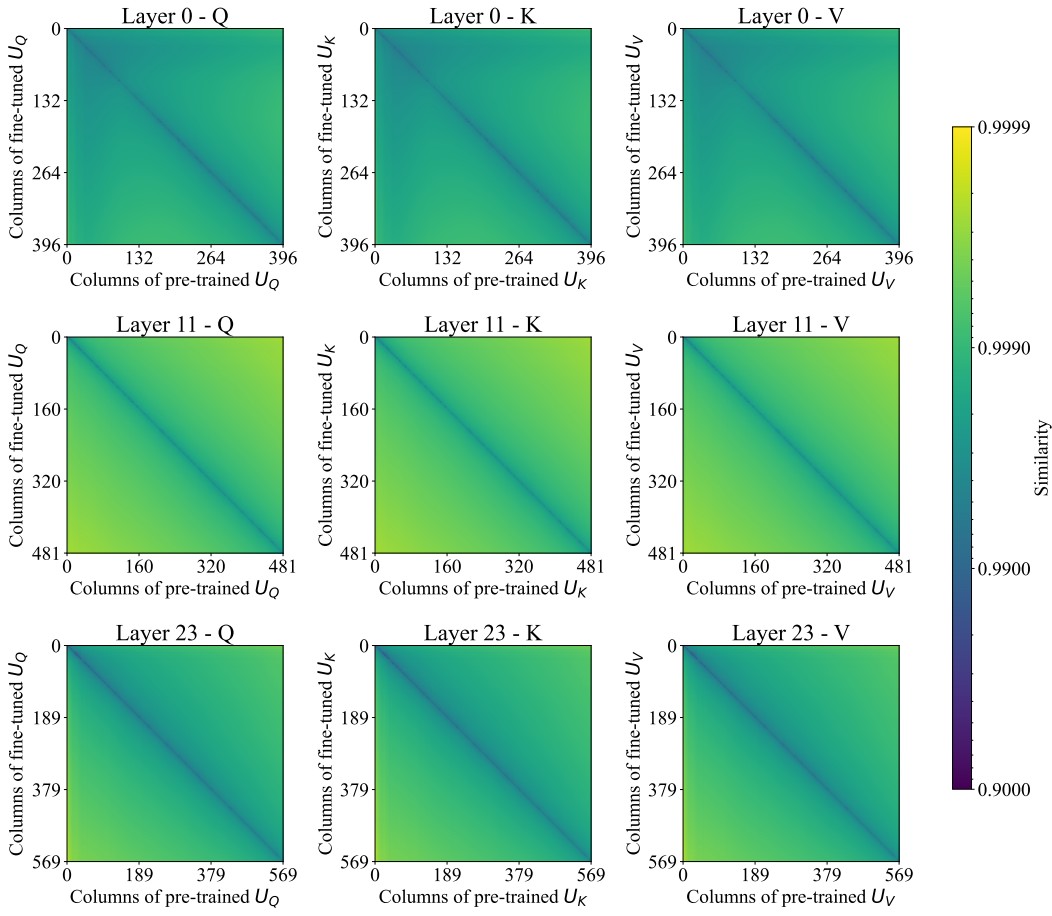

Figure 4: The similarity between the principal output subspace $U_p$ of the pre-trained and fine-tuned weights for the Q, K, and V matrices in layers 0, 11, and 23 of ViT-Large (Dosovitskiy, 2021). The x-axis represents the subspace spanned by the top-$i$ singular vectors of the pre-trained weights, while the y-axis represents the subspace spanned by the top-$j$ singular vectors of the fine-tuned weights.

# F SUBSPACE SIMILARITY ANALYSIS

## F.1 SUBSPACE SIMILARITY BETWEEN PRE-TRAINED AND FINE-TUNED MODELS

Our theoretical analysis assumes that CERSA can approximate full-parameter fine-tuning based on the premise that the principal subspace of $W'$ after full fine-tuning on a downstream task remains highly similar to that of the pre-trained weights. This assumption suggests that fine-tuning primarily refines the existing subspace rather than significantly altering its structure. To empirically validate

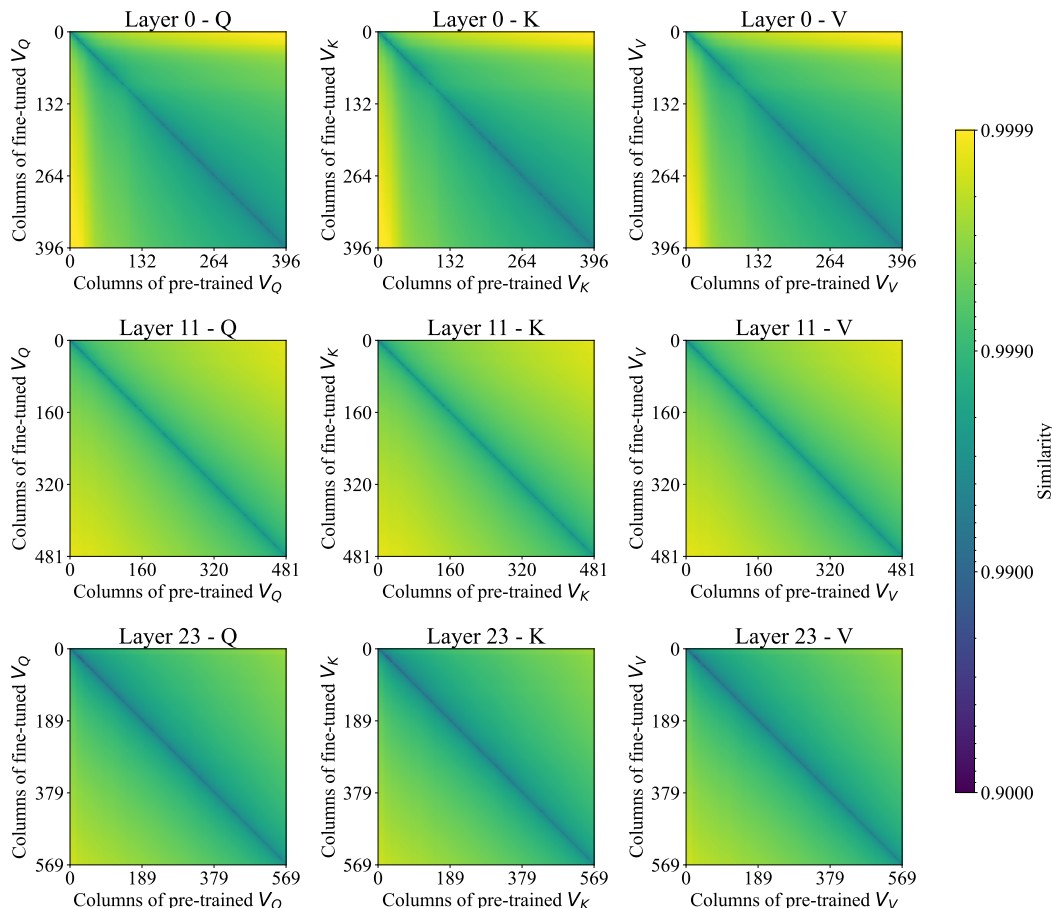

Figure 5: The similarity between the principal input subspace $\boldsymbol{V}_p$ of the pre-trained and fine-tuned weights for the Q, K, and V matrices in layers 0, 11, and 23 of ViT-Large (Dosovitskiy, 2021). The x-axis represents the subspace spanned by the top-$i$ singular vectors of the pre-trained weights, while the y-axis represents the subspace spanned by the top-$j$ singular vectors of the fine-tuned weights.

this assumption, we measure the subspace similarity between $\boldsymbol{W}'$ and the pre-trained weights $\boldsymbol{W}$ of the ViT-Large (Dosovitskiy, 2021) model, initially trained on ImageNet-21K (Deng et al., 2009), across eight different downstream image classification datasets.

To quantitatively assess this similarity, we employ the *Grassmann subspace similarity* (Hu et al., 2022), a metric that effectively captures the alignment between the principal output subspaces of the pre-trained and fine-tuned weights. Formally, the Grassmann similarity is defined as follows:

$$\psi(\boldsymbol{U}_A^i, \boldsymbol{U}_B^j) = \frac{\|\boldsymbol{U}_A^{i\top}\boldsymbol{U}_B^j\|_F^2}{\min\{i, j\}}, \quad \psi \in [0, 1] \tag{1}$$

where $\boldsymbol{U}_A^i$ and $\boldsymbol{U}_B^j$ stands for top-$i$ columns of the $\boldsymbol{U}$ matrix from SVD decomposition of matrix $\boldsymbol{A}$ and top-$j$ columns of the $\boldsymbol{U}$ matrix from SVD decomposition of matrix $\boldsymbol{B}$ respectively. $\psi$ ranges from 0 (completely disjoint subspaces) to 1 (identical subspaces).

Similarly, we extend this analysis to the input subspace represented by $\boldsymbol{V}$, applying the same similarity computation. The results presented in Tab. 8 are computed using a subspace that retains 95% of the cumulative energy. The first value represents the similarity of the output subspace $\boldsymbol{U}$, while the second corresponds to the input subspace $\boldsymbol{V}$.

Analyzing Tab. 8, we observe that across all downstream tasks, the Grassmann subspace similarity between the fine-tuned and pre-trained subspaces consistently exceeds 99.5% for both $\boldsymbol{U}$ and $\boldsymbol{V}$

across all three attention matrices – Q, K, and V. This strong evidence suggests that fine-tuning minimally affects the principal subspace of the pre-trained weights, thereby validating our assumption.

To further examine the stability of the Grassmann subspace similarity under varying top-$k$ selections, we conducted experiments on the 0th, 11th, and 23rd layers of the ViT-Large (Dosovitskiy, 2021) model before and after fine-tuning on CIFAR-10 (Krizhevsky & Hinton, 2009). Specifically, we extract the Q, K, and V matrices from these layers, perform SVD to obtain the principal subspaces $U$ and $V$ for both the pre-trained and fine-tuned models, and measure subspace similarity by selecting top-$i$ and top-$j$ singular vectors. The results are visualized in a heat map.

Since all similarity values fall within the range of 0.9 to 0.9999, we applied a logarithmic transformation to the color scale for better visualization. As depicted in Fig. 4 and Fig. 5, the subspaces of the fine-tuned and pre-trained weight matrices exhibit consistently high similarity across all choices of top-$i$ and top-$j$, with values ranging from 0.9 to 0.9999. This confirms that the observed subspace similarity is not confined to specific top-$k$ selections but persists across all choices of singular vectors.

Regardless of the truncation level, the fine-tuned and pre-trained weight matrices maintain exceptionally high Grassmann subspace similarity. This finding further substantiates our hypothesis that fine-tuning does not significantly alter the principal subspace of the pre-trained model, reinforcing the fundamental assumption underlying our method.

## F.2 PROOFS

*Proof.* Let $M \in \mathbb{R}^{m \times n}$ be a matrix of rank $k$. By the singular value decomposition (SVD), we can write $M = U\Sigma V^T$, where $U \in \mathbb{R}^{m \times k}$ and $V \in \mathbb{R}^{n \times k}$ are matrices with orthonormal columns, and $\Sigma \in \mathbb{R}^{k \times k}$ is a diagonal matrix with positive diagonal entries (the singular values).

Since there exists an orthonormal basis $Q = \{e_1, e_2, \ldots, e_k\}$ such that $\mathrm{Span}(U) = \mathrm{Span}(Q)$, both $U$ and $Q$ form orthonormal bases for the same $k$-dimensional subspace. Therefore, there exists an orthogonal matrix $R \in \mathbb{R}^{k \times k}$ such that
$$U = QR.$$

Similarly, because $\mathrm{Span}(V) = \mathrm{Span}(Q')$, there exists an orthogonal matrix $R' \in \mathbb{R}^{k \times k}$ satisfying
$$V = Q'R'.$$

Substituting these expressions into the singular value decomposition of $M$, we obtain
$$M = U\Sigma V^T = (QR)\Sigma(Q'R')^T = QR\Sigma R'^T Q'^T.$$

Defining $S = R\Sigma R'^T$, we have
$$M = QSQ'^T,$$
where $S \in \mathbb{R}^{k \times k}$. This completes the proof. $\square$