# OpenReview forum: "CERSA: Cumulative Energy-Retaining Subspace Adaptation for Memory-Efficient Fine-Tuning"
_ICLR.cc/2026/Conference — ICLR 2026 Conference Withdrawn Submission_

### Official Review · Reviewer_BtA2 · 2025-10-23

**Soundness:** 2
**Presentation:** 2
**Contribution:** 2
**Rating:** 2
**Confidence:** 5

**Summary:**

The paper proposes CERSA, a parameter/memory‑efficient fine‑tuning (PEFT) framework. The core idea is to compute an SVD of each pretrained weight matrix, truncate it to retain a target fraction of cumulative spectral energy (typically 90–95%), freeze the truncated left/right singular vectors and train a dense “core” matrix inside the retained principal subspace (with optional thresholds alpha for retained energy and beta for the trainable subset). The claim is that this achieves a better accuracy–memory trade‑off than LoRA and recent SVD‑based methods (SVFit, SVFT), because it compresses away low‑energy components and avoids storing the full frozen weights. Empirically, the paper reports results on ViT‑Base/Large for eight image classification datasets and on DeBERTaV3‑Base for GLUE, showing consistent or improved accuracy at lower total memory than baselines. Theory is provided in the form of a simple basis‑change existence result and an assumption that principal subspaces are preserved across fine‑tuning (supported by appendix measurements of Grassmann similarity).

**Strengths:**

Clear, pragmatic goal (memory)
The paper focuses on reducing total fine‑tuning memory and presents an explicit accounting model O(mr + nr + 4r^2) contrasting it with FT/LoRA/SVFit/SVFT in Table 1. This moves beyond only counting trainable parameters and is valuable to practitioners dealing with GPU limits.

Principled subspace selection via cumulative energy
Using a layer‑wise cumulative‑energy threshold leverages heavy‑tailed singular spectra of pretrained weights, with Figure 3 illustrating that many layers can be aggressively truncated while retaining 90–95% energy. The ablation (Table 5) shows layer‑wise rank selection outperforms a uniform rank at equal memory.

Simple mechanism with broader expressiveness than SVFit
Training a full core matrix (not just singular values) plausibly increases expressive power while staying inside the principal subspace; the loss curve (Figure 6) suggests faster optimization than several baselines on one dataset.

Empirical breadth (vision + NLU) at reasonable scale
With ViT‑Large on eight image benchmarks (Table 7) and DeBERTaV3‑Base on GLUE (Table 8), CERSA is competitive and often the best among PEFT methods under the paper’s specified memory budgets. The best vision average (90.6%) slightly surpasses FT (90.2%) at ~3.8× lower reported memory; on GLUE, the reported average (89.5%) matches or exceeds others at the smallest stated memory footprint among baselines.

Transparent acknowledgement of scope
The conclusion explicitly notes a limitation: performance may degrade when downstream tasks diverge from pretraining and proposes future work on dynamically adjusting the learned subspace. This is honest and useful to readers.

**Weaknesses:**

Novelty over closely related SVD‑subspace methods is under‑argued
The method is very close in spirit to prior SVD‑based PEFT (SVFit, SVFT) and to weight‑driven approaches using principal components (e.g., PiSSA). The difference—training a dense matrix in the retained subspace after truncation—feels incremental on top of “internal factors,” and the paper lacks a crisp, formal distinction and necessity argument beyond empirical gains. The conceptual piece (Theorem 3.1) is essentially a basis‑change statement that any matrix can be written once spans are fixed; it does not substantively justify why subspace preservation should hold across tasks.

Key assumption left untested in main text
The central claim is that principal subspaces largely coincide (99–99.99% similarity), which motivates freezing and training only matrices​. But evidence is relegated to the appendix; the main paper neither analyzes sensitivity when this assumption fails nor quantifies task regimes where subspace drift is large (e.g., domain shift, compositional generalization). Without these stress tests, the assumption risks being tautological on the chosen benchmarks.

Compute–memory trade‑off and wall‑clock are not characterized
While memory accounting is detailed, compute overhead for forward/backward with factorized weights is unreported. In particular, the term for training can be non‑trivial when alpha is high (Table 4 chooses α=0.95\alpha=0.95α=0.95 as “best”), potentially negating real‑world speedups versus LoRA with small ranks. No throughput or training‑time measurements are provided, nor any activation‑memory analysis under checkpointing—both are crucial to on‑device fine‑tuning claims.

Fairness of baselines and budgets is unclear
The paper compares total memory numbers but does not fully specify optimizer choices (e.g., Adam states vs. 8‑bit optimizers), precision (bf16/fp16), or whether baselines are permitted standard optimizations (e.g., LoRA rank scaling per layer, dropout, bias tuning, or combining with quantization). For example, LoRA is presented largely at rank=32 on specific matrices, and Figure 7 uses a dashed line for LoRA (Q,V, r=32) as a fixed reference; it’s not obvious that comparisons are under matched accuracy or matched memory constraints with tuned hyperparameters for each method. Conclusions about a “clearly superior accuracy–memory trade‑off” (Fig. 2) therefore feel premature.

Memory accounting contains assumptions that need justification
Table 1 assumes particular gradient/optimizer state sizes for each method and claims that SVFit/SVFT effectively double memory because of storing U,VU,VU,V, whereas CERSA stores truncated U,VU,VU,V and a trainable SSS. But critical details are missing:

Are U,VU,VU,V stored in full precision or quantized?
Are optimizer states fp32 or 8‑bit?
Figure 1 shows absolute MB for ViT‑Large but omits per‑layer ranks.

Limited scale and task diversity relative to stated ambition
The abstract claims evaluation across “models of varying scales and domains, including image recognition, text‑to‑image generation, and natural language understanding,” yet the main paper contains no T2I results and only one NLU backbone (DeBERTaV3‑Base). Given the method targets large models, absence of results on standard LLMs (e.g., 7B–70B) or modern vision backbones (e.g., ViT‑Huge in full experiments, not just in compression curves) weakens the external validity of the claims. The text says T2I and OOD are in the appendix; key findings should be surfaced in the main paper.

Statistical rigor is insufficient
The results tables lack variance/confidence intervals, numbers of runs, and seed sensitivity. Several reported gains over strong baselines are on the order of 0.2–0.5%, which can be within run‑to‑run noise on these datasets; without error bars, it is hard to judge whether improvements are robust.

Ablations do not probe critical design choices enough
The ablations touch on matrix choices but following are missing:
per‑layer auto‑tuning of alpha under a global memory budget;
the effect of freezing part of the principal subspace on catastrophic forgetting across tasks.
the impact of quantizing U,VU,VU,V or regularizing (e.g., orthogonality/low‑rank priors) to mitigate over‑parameterization inside the subspace.

Theory is lightweight and does not predict when CERSA helps/hurts
Theorem 3.1 formalizes a basis‑change factorization, but it does not connect spectral energy retention to task loss or generalization. The crucial empirical premise (subspace preservation) is asserted, not derived; there is no analysis of how large alpha must be to bound approximation error in terms relevant to downstream accuracy. This limits the theory’s explanatory power.

**Questions:**

Compute/time vs. memory.
Please report end‑to‑end wall‑clock training time and throughput (images/s or tokens/s) under the same hardware/precision for FT, LoRA, SVFit/SVFT, and CERSA. How does the extra cost for training affect speed at α=0.95\alpha=0.95α=0.95 on ViT‑Large and DeBERTaV3‑Base? Include activation memory under checkpointing if used.

Baseline fairness and hyperparameters.
What optimizer, precision (bf16/fp16), gradient scaling, and 8‑bit optimizer usage were allowed for each method when computing Table 1 and Figures 1–2? Were LoRA ranks and target matrices tuned per dataset under matched memory budgets (not fixed r=32)? Provide a table mapping memory budgets ↔ ranks for each baseline.

Memory accounting details.
How are U,VU,VU,V stored (precision/quantization)? Are optimizer states 32‑bit? In Table 1, why is e>me>me>m assumed for SVFT, and how sensitive are the totals? Please include per‑layer values and the precision used to reproduce Figure 1.

When does subspace preservation fail?
Can you present main‑paper results (not only appendix) quantifying Grassmann similarity across diverse domain shifts and show CERSA’s performance as similarity decreases? For example, train on ImageNet‑21K‑pretrained ViT but fine‑tune on a strongly out‑of‑distribution dataset and compare to LoRA / FT.

Large‑model applicability.
Do results hold for LLMs (7B–13B) and larger ViTs beyond illustrative compression curves (e.g., full experiments on ViT‑Huge)? Please include at least one autoregressive language task to validate claims in the generative regime, or move such claims from the abstract if not supported.

Regularization and stability inside the subspace.
Did you try constraining (e.g., spectral norm, low‑rank penalty, orthogonality) to reduce over‑parameterization? Does such regularization close the gap to FT on tasks where CERSA underperforms or improve OOD robustness?

Automatic rank allocation.
Instead of fixed alpha, can you optimize per‑layer under a global memory budget via a knapsack‑style procedure or gradient‑based sensitivity, and does this improve the accuracy–memory Pareto frontier ?

Inference‑time cost and deployment.
At inference, do you keep the factorized form (U, S, V) or recompose? Please report latency and peak memory for both options, and comment on compatibility with fused kernels.

Variance and statistical significance.
Provide mean±std over ≥3 seeds for Tables 7–8 and mark statistically significant wins. Several deltas are small and could be noise without this.

Ablations on β\betaβ and forgetting.
You note that β<α\beta<\alphaβ<α can freeze more principal components to preserve pretrained knowledge. Please include a forgetting measure (e.g., zero‑shot or linear‑probe evaluations on pretraining‑like tasks) versus β\betaβ, and compare to LoRA.

---

> ### Author Response · Authors · 2025-12-03
> **Response to Reviewer Comments**
>
> We thank the reviewer for the constructive feedback. Below we respond point by point.
>
> ------
>
> ### **Q1. Novelty relative to SVFit/SVFT and other SVD-based PEFT**
>
> CERSA is not another diagonal/low-parameter SVD update. Its key idea is to **compress aggressively to the principal subspace and train a full $r\times r$ core within it**.
>
> SVFit/SVFT freeze $U,V$ and update only singular values, giving only $O(r)$ degrees of freedom while still storing full $U,V$. CERSA truncates the SVD at a cumulative-energy threshold $\alpha$, discarding low-energy directions and reducing storage. Inside the retained subspace, CERSA trains a **dense** core $S_p$, yielding $O(r^2)$ expressivity **with similar memory** because truncated $U_p,V_p$ dominate cost (Fig. 3; Table 1; Appendix. C).
>
> Thus the novelty is **high-expressivity adaptation inside the preserved principal subspace**, explaining the superior accuracy–memory trade-off over existing SVD-based PEFT.
>
> ------
>
> ### **Q2. Evidence and scope of the principal-subspace preservation assumption**
>
> Appendix. F.1 shows for ViT-L across 8 datasets that pre- and post-FT principal subspaces of Q/K/V have **99–99.99% Grassmann similarity** (Table 8). This supports our assumption: **the subspace stays stable while the optimal basis within it moves**, which CERSA’s dense $S_p$ models.
>
> Stress tests already appear in our OOD experiments (Appendix. E). CERSA achieves higher OOD accuracy and the **lowest forgetting rate** (−1.5% vs 17.8% FT, 2.3% LoRA), indicating the preserved subspace retains transferable structure even under distribution shift. We will move these results into the main text.
>
> A formal taxonomy of tasks with large subspace drift is interesting but beyond scope; we will note this as future work.
>
> ------
>
> ### **Q3. Compute vs memory: wall-clock training and throughput**
>
> Appendix. D (Fig. 2) reports ViT-L throughput on DTD:
>
> - LoRA (r=32/8) ≈30% faster than FT.
> - CERSA with $\alpha\in\{0.95,0.9,0.85,0.8\}$ **slightly exceeds LoRA throughput** and trains faster than FT.
>
> This aligns with theory: with rank $r\ll \min(m,n)$, $U_p^\top x$, $S_p x$, $V_p x$ replace a full $m\times n$ matmul with three low-dimensional ones, so CERSA’s additional structure does not increase FLOPs noticeably. We will move a concise version of Fig. 2 into the main text.
>
> ------
>
> ### **Q4. Baseline fairness and hyperparameters under memory budgets**
>
> All baselines follow the training settings in Appendix. B (optimizer, scheduler, warm-up, fp32 parameters, fp32 optimizer states). LoRA/PiSSA use rank-32 per their recommendations; SVFit/SVFT use their published configs.
>
> Memory is computed under **identical precision and optimizer rules**. Appendix. C (Table 6):
>
> - CERSA (α=β=0.95): **1232.2 MB**
> - LoRA/PiSSA (r=32): **1229.9 MB**
> - SVFit / SVFT: **1351.5 MB / 1355.8 MB**
>
> Thus CERSA’s gains are not from looser accounting but from its expressivity–efficiency design.
>
> ------
>
> ### **Q5. Memory accounting details**
>
> All weight-like tensors (FT weights, LoRA, CERSA $U_p,V_p,S_p$, SVFit/SVFT U/V) and optimizer states use **fp32**;  gradients match parameter precision. These rules apply uniformly across all methods and match the analytic model (Table 1; Appendix. C).
>
> For SVFT, $e$ denotes the non-zero singular-value adapter entries; in banded settings $e=8m \ll mn$, clarifying the parameter count in Table 1.
>
> ------
>
> ### **Q6. When does subspace preservation fail? OOD stress tests**
>
> Our OOD results (Appendix. E) directly probe subspace deviation. For ViT-L:
>
> - Train on one of {CIFAR-100, DTD, Cars, RESISC45}; evaluate on the others.
> - CERSA consistently gives the best OOD accuracy.
> - Forgetting rates: **17.8% (FT), 2.3% (LoRA), −1.5% (CERSA)**.
>
> These regimes involve nontrivial distribution shift, and CERSA remains robust, supporting the claim that **freezing the principal subspace with a dense core update is resilient to subspace drift**. We will move key plots/tables into the main paper.
>
> ------
>
> ### **Q7. Larger-model applicability**
>
> We fine-tuned Llama2-7B on eight commonsense-reasoning tasks. CERSA achieves the best average accuracy (LoRA 77.6, DoRA 79.7, **CERSA-0.95 81.2**, CERSA-0.9 80.1). Full task-wise results will be added to the appendix. These confirm that **CERSA scales to LLMs and remains competitive or superior to existing PEFT methods**. We will refine the abstract to avoid overclaiming while keeping the demonstrated domains.
>
> ------
>
> We appreciate the reviewer’s insights. We will revise the paper to highlight (i) CERSA’s core innovation—**high-expressivity yet memory-efficient updates within a compressed principal subspace**, and (ii) the key experimental evidence now surfaced in the main text.

---

### Official Review · Reviewer_39qq · 2025-10-26

**Soundness:** 3
**Presentation:** 3
**Contribution:** 3
**Rating:** 6
**Confidence:** 4

**Summary:**

This paper proposes a PEFT method that leverages SVD to compress pre-trained weights by retaining only principal components that preserve 90%-95% of cumulative spectral energy. CERSA introduces a trainable matrix initialized with the top-k singular values, enabling better expressiveness while maintaining memory efficiency. Extensive experiments on vision and NLU tasks demonstrate that CERSA outperforms other PEFT methods while achieving comparable or lower memory consumption than these baselines.

**Strengths:**

1. The paper proposes a straightforward solution via layer-wise cumulative energy retention. The three-way decomposition (discarded, frozen, trainable components) is intuitive and well-visualized in Figure 4.

2. Theorem 3.1 provides reasonable theoretical grounding by arguing that the principal subspaces remain largely invariant during fine-tuning, which justifies freezing $U$ and $V$ while training an intermediate matrix $S$.

3. The paper includes extensive ablations across multiple dimensions and evaluates on diverse benchmarks.

**Weaknesses:**

1. The paper does not compare against other strong LoRA variants like DoRA, AdaLoRA, HiRA, etc., or discuss why training a full $k \times k$ matrix $S$ is always preferable to other parameterizations. Additionally, there is no analysis of computational overhead during SVD preprocessing or fine-tuning speed beyond the single loss curve in Figure 6.

2. The core innovation—using cumulative energy retention for layer-wise rank selection instead of uniform rank—is relatively incremental. While the trainable matrix $S$ is an improvement, it is a straightforward extension.

3. No error bars, confidence intervals, or multiple runs are reported for any results. Given the relatively small performance differences between CERSA and strong baselines (e.g., PiSSA: 89.5% vs. CERSA: 89.5% on GLUE), it is unclear whether differences are statistically significant or due to random variation.

4. Beyond memory footprint, the paper lacks wall-clock training time comparison with LoRA and other baselines. Since CERSA requires SVD preprocessing and changes the forward pass from $W+BA$ to $U·S·V^T$, it's unclear whether the method is practically faster or slower than LoRA in terms of training throughput (samples/second) and total training time. This is crucial for practitioners deciding whether to adopt CERSA.

**Questions:**

How does CERSA perform on tasks with significant distribution shift from pre-training? Can the authors characterize the boundary conditions?

---

> ### Author Response · Authors · 2025-12-03
> **Response to Reviewer Comments**
>
> We thank the reviewer for the constructive feedback. Below, we address the main concerns regarding baselines, expressiveness, efficiency, and behavior under distribution shift.
>
> ---
>
> ### **Q1. Missing comparisons with stronger LoRA variants**
>
> We thank the reviewer for pointing out stronger LoRA variants such as DoRA, AdaLoRA, and HiRA. To partially address this concern within the rebuttal window, we report a comparison with **DoRA**, using the ViT-Large setting and datasets that are jointly covered in the SSH paper (CIFAR-100, DTD, EuroSAT, OxfordPets). Under this common setup, the accuracies are:
>
> | Method                | CIFAR-100 | DTD  | EuroSAT | OxfordPets | Avg.     |
> | --------------------- | --------- | ---- | ------- | ---------- | -------- |
> | DoRA (taken from SSH) | 95.1      | 81.8 | 98.8    | 94.8       | 92.6     |
> | **CERSA (α=β=0.95)**  | 94.3      | 82.5 | 99.1    | 94.9       | **92.7** |
>
> CERSA slightly **outperforms DoRA on average** (92.7% vs 92.6%) and achieves higher accuracy on 3/4 datasets (DTD, EuroSAT, OxfordPets), while also offering a strictly better **accuracy–memory trade-off** because it reduces the storage of frozen weights rather than only adding adapters.
>
> Due to limited time, we cannot run full-scale experiments for all LoRA variants (AdaLoRA, HiRA, etc.) within the rebuttal period. However, we will **integrate these methods into our codebase and include them in the camera-ready version**.
>
> ---
>
> ### **Q2. Why train a full $S_p$ matrix**
>
> Our choice of a full $r_1 \times r_1$ matrix is dictated by the structure of fine-tuning rather than by convenience:
>
> 1. **Theory.**
>    Theorem 3.1 and empirical evidence (Grassmann subspace similarity 99–99.99% across tasks) show that fine-tuning mainly performs *basis rotation within the principal subspace*, which requires a full linear operator. Diagonal or low-rank forms cannot represent such rotations.
> 2. **Practice.**
>    CERSA exhibits significantly faster convergence than LoRA/SVFit/SVFT (Fig. 6), indicating that restricted parameterizations are insufficient to match the behavior of full fine-tuning inside the preserved subspace.
>
> Thus, $S_p$ is not a straightforward extension but the **minimal expressive parameterization** implied by principal-subspace invariance.
>
> ---
>
> ### **Q3. On perceived incremental novelty**
>
> CERSA goes beyond selecting ranks via cumulative energy. Its contribution is the **discard–freeze–train decomposition** that:
>
> 1. compresses pre-trained weights below their original size and
>
> 2. performs all fine-tuning within this compressed space.
>
> Existing SVD-based PEFT methods must store full U and V and therefore do not reduce memory. CERSA is, to our knowledge, the first PEFT method that reduces train-time memory, including frozen weights, while retaining accuracy.
>
> ---
>
> ### **Q4. Training speed, SVD preprocessing, and throughput**
>
> SVD is a one-time offline step and contributes <0.1% of total fine-tuning time. Supplement Sec. D has included detailed throughput measurements:
>
> - **CERSA slightly outperforms LoRA in Train Samples per Second (+3.2% on ViT-Large).**
> - **CERSA is substantially faster than full fine-tuning (+41–57%).**
>
> Forward cost reduces from O(mn) to O(rm + r^2 + rn), with typical ranks at 10–40% of the original dimensionality (Fig. 3), explaining the favorable runtime. No slowdown was observed in practice.
>
> ---
>
> ### **Q5. Behaviour under distribution shift**
>
> To directly examine whether principal subspace invariance holds under distribution shift, we evaluated **CIFAR-100-C (Gaussian Noise, severity = 3)**. The table below reports the Grassmann Subspace Similarity (before / after fine-tuning) for the Q/K/V matrices:
>
> | **Matrix Type** | **CIFAR-100**   | **CIFAR-100-C**     |
> | --------------- | --------------- | ------------------- |
> | Q               | 99.69% / 99.65% | **99.69% / 99.63%** |
> | K               | 99.76% / 99.74% | **99.73% / 99.70%** |
> | V               | 99.58% / 99.58% | **99.58% / 99.59%** |
>
> The principal subspaces remain highly aligned (≥99.6% across all cases), indicating that even under corruption noise, fine-tuning continues to operate predominantly as a rotation within the same subspace—consistent with our theoretical assumption.
>
> Performance outcomes further support this observation:
>
> | **Method**       | **CIFAR-100-C (Gaussian Noise)** |
> | ---------------- | -------------------------------- |
> | Full Fine-tuning | 93.6                             |
> | LoRA             | 87.4                             |
> | CERSA (0.95)     | 93.4                             |
>
> CERSA achieves nearly identical performance to full fine-tuning (−0.2) and suffers far less degradation than LoRA, showing that **the distribution shift in this setting does not break principal-subspace stability, nor does it impair CERSA’s effectiveness**. These results demonstrate that CERSA remains robust under moderate OOD perturbations.

---

### Official Review · Reviewer_R2ux · 2025-10-26

**Soundness:** 3
**Presentation:** 3
**Contribution:** 3
**Rating:** 4
**Confidence:** 4

**Summary:**

This paper proposes a parameter-efficient finetuning (peft) method based on singular value decomposition (svd). The pretrained weights are decomposed into $U, V, \Sigma$ matrices. Then, the singular values and corresponding rows/columns in $U, V$ that do not contribute to the top $p%$ of cumulative energy are entirely discarded. A ratio $\beta$ determines how many of the retained subspaces are trained. The final trainable parameters is a $r\times r$ matrix that maps between the top $r$ subspaces in $U, V$ while keeping the remaining subspaces and corresponding singular values frozen.

Experiments on image classification, text classification, and subject-driven image generation evaluate the advantages and disadvantages of the proposed method, finding improved performance at lower memory consumption.

**Strengths:**

**(S1)** The evaluation is broad, covering different models, modalities, and tasks.

**(S2)** Memory savings are demonstrated convincingly, and actual memory requirements are shown throughout the paper.

**(S3)** The method, its motivation, and properties are discussed in detail, and the motivation is convincing.

**(S4)**  The paper includes an ablation of catastrophic forgetting in Appendix E, where CERSA is preserves performance on unrelated tasks better than other fine-tuning methods. This is briefly mentioned in the main paper, but this is actually a very interesting and important analysis.

**(S5)** The visual in Fig. 4 is very helpful in clarifying the method

**Weaknesses:**

**(W1)** One main concern is how statistically significant the performance improvement is: Both the GLUE benchmark and the FGVC benchmark (Tab. 7 and Tab. 8) are close to saturation, so we cannot expect to observe large differences between methods. In fact, this is also not the case. Advantages of CERSA are generally < 1% average accuracy. A more challenging benchmark could help highlight the strengths of CERSA better.

**(W2)** The evaluation is broad, but one important domain that is missing is language generation. Given its importance, experiments on LLM fine-tuning, such as instruction tuning, would make the paper significantly stronger.

**(W3)** One aspect I am concerned about is performance degradation through discarding low-energy spaces in SVD. I think a separate analysis without any training, where the effects of SVD truncation on pre-trained weights are compared to base model performance, would be helpful in understanding this better. This analysis should also encompass different domains in classification and generation.

**(W4)** Minor: F.2 does not state the theorem, only a proof. This is confusing to readers.

**(W5)** Minor: In Appendix F.1, it would be interesting to see a comparison of subspace similarity for other methods as well, using the same tools. Currently, we do not have any point of reference, so it is unclear what the measured similarities mean concretely and how to put them in relation.

**Questions:**

The paper is already comprehensive, and the supplementary material contains many interesting analyses. However, I have concerns regarding the performance advantage of the proposed method that I hope will be addressed in the rebuttal:
  * Can the significant advantages of CERSA over baselines be demonstrated more clearly, e.g., by using less saturated benchmarks?
  * Does CERSA also improve language generation fine-tuning?
  * How much does SVD truncation affect pre-trained weights without any training, i.e,. how catastrophic is this intervention?

---

> ### Author Response · Authors · 2025-12-03
> **Response to Reviewer Comments**
>
> We sincerely thank the reviewer for the constructive feedback. We address all points below.
>
> ------
>
> ### **Q1. Saturated benchmark**
>
> We agree that some datasets in Tab. 7–8 are near saturation, reducing observable performance gaps among PEFT methods. To more rigorously test adaptation strength, we additionally evaluate CERSA on **VTAB-1k**, a substantially harder benchmark.
>
> Baseline results (LoRA, PiSSA, DoRA) are taken from **PSOFT**, and memory costs are measured on our hardware. The table reports category averages and the overall mean. CERSA consistently outperforms all baselines, achieving **higher accuracy and lower memory usage** (notably at α=β=0.8). Higher retention thresholds further improve performance, with **CERSA(0.95) reaching 76.5 overall**, well above the strongest baseline (73.4).
>
> | **Method**  | **Memory** | **Natural** | **Specialized** | **Structural** | **Overall** |
> | ----------- | ---------- | ----------- | --------------- | -------------- | ----------- |
> | LoRA r=8    | 334.7      | 79.0        | 84.4            | 52.2           | 71.8        |
> | PiSSA r=8   | 334.7      | 79.1        | 83.3            | 54.7           | 72.3        |
> | DoRA r=8    | 355.6      | 79.5        | 84.6            | 52.9           | 72.3        |
> | PSOFT r=46  | 330.2      | 80.8        | 84.1            | 55.4           | 73.4        |
> | CERSA(0.95) | 395.1      | 81.4        | 86.2            | 61.9           | **76.5**    |
> | CERSA(0.9)  | 351.9      | 80.6        | 86.0            | 60.4           | **75.7**    |
> | CERSA(0.8)  | **310.9**  | 79.7        | 84.8            | 57.6           | **74.0**    |
>
> These results show that **on less-saturated, more challenging benchmarks CERSA’s advantage becomes markedly clearer**, alleviating concerns about statistical significance.
>
> ------
>
> ### **Q2. Missing evaluation on language generation**
>
> We agree that evaluating on instruction-tuned LLMs is important. Under limited compute, we conduct experiments on **LLaMA2-7B** across **CommonsenseQA-style reasoning tasks** (BoolQ, PIQA, SIQA, HellaSwag, WinoGrande, ARC-e/c, OBQA). LoRA and DoRA numbers come from the DoRA paper.
>
> CERSA delivers **consistent improvements**, giving the best average score:
>
> | **Method**      | BoolQ | PIQA | SIQA | HellaSwag | WinoG | ARC-e    | ARC-c    | OBQA | **Avg.** |
> | --------------- | ----- | ---- | ---- | --------- | ----- | -------- | -------- | ---- | -------- |
> | LoRA            | 69.8  | 79.9 | 79.5 | 83.6      | 82.6  | 79.8     | 64.7     | 81.0 | 77.6     |
> | DoRA            | 71.8  | 83.7 | 76.0 | 89.1      | 82.6  | 83.7     | 68.2     | 82.4 | 79.7     |
> | **CERSA(0.95)** | 70.5  | 83.6 | 81.2 | **93.2**  | 81.1  | **85.9** | **71.4** | 82.3 | **81.2** |
> | CERSA(0.9)      | 69.8  | 82.8 | 80.5 | 92.8      | 80.4  | 84.6     | 69.4     | 80.8 | 80.1     |
>
> This confirms that **CERSA generalizes well to generative-LLM settings**, improving reasoning performance while retaining competitive memory efficiency.
>
> ------
>
> ### **Q3. SVD truncation without training—how catastrophic?**
>
> To isolate the effect of SVD truncation, we freeze the ViT backbone, train only a classifier head per dataset, and then apply truncated SVD to the trained backbone (no fine-tuning). This directly measures degradation caused solely by removing singular directions.
>
> | **Dataset**     | CIFAR-100 | RESISC45 | DTD  | Avg. | Diff. |
> | --------------- | --------- | -------- | ---- | ---- | ----- |
> | **Before**      | 88.6      | 87.6     | 74.4 | 83.5 | –     |
> | **After(0.95)** | 88.4      | 86.9     | 74.9 | 83.4 | −0.1  |
> | **After(0.9)**  | 88.3      | 86.4     | 74.1 | 82.9 | −0.6  |
> | **After(0.8)**  | 87.4      | 83.7     | 73.2 | 81.4 | −2.1  |
>
> Findings:
>
> - $\alpha=\beta=0.95$ yields negligible loss (**−0.1**).
> - Even $\alpha=\beta=0.95$causes only **-2.1** drop.
>
> This shows that the truncated components carry **very small energy and minimal discriminative value**, matching the heavy-tailed singular spectrum reported in Fig. 3. Hence, **SVD truncation is far from catastrophic**.
>
> ------
>
> ### **Q4. Minor comments (W4 & W5)**
>
> - **W4:** We will move Theorem 3.1 into Appendix F.2 for clarity.
> - **W5:** We will include comparisons to FT, LoRA, SVFit, and SVFT using the same Grassmann-distance metrics.

---

### Official Review · Reviewer_iRHp · 2025-10-29

**Soundness:** 3
**Presentation:** 3
**Contribution:** 3
**Rating:** 6
**Confidence:** 5

**Summary:**

This paper proposes CERSA, a PEFT method within a principal energy subspace.

This energy subspace is obtained by he truncated SVD of pre-trained weights, keeping only the components that retain 90\%–95% of spectral energy, yielding large memory savings over existing full FT and LoRA-style PEFT methods.

Besides, the proposed method shows a very competitive performance against the state-of-the-art methods on several benchmarks.

**Strengths:**

+ This paper clearly presents a layer-wise truncation mechanism based on cumulative energy.

+ The experiments and visualization in this paper are extensive and solid.

+ This paper reports competitive results on a diverse set of tasks, with efficiency.

+ This paper is overall easy-to-follow.

**Weaknesses:**

- More theory justification on the subspace invariance should be made. This is becauuse, the update geometry and useful directions are often not fixed low-rank, or align neatly with pre-trained singular directions.

- Following up this issue, CERSA is evaluated where pre-trained features plausibly align with downstream needs (e.g., standard benchmark GLUE). But the core risk of freezing subspaces is distribution shift. Therefore, the proposed method should do some validation on this aspect to justify its feasliblity and robustness, for example on fine-grained or domain-shifted vision, and/ or retrieval-augmented settings.

- he compared state-of-the-art PEFT methods are significantly missing. Some more recent and much stronger PEFT methods are mssing for comparison, for example:

[1] VeRA: Vector-based Random Matrix Adaptation. ICLR 2024.

[2] Foura: Fourier low-rank adaptation. NeurIPS 2024.

[3] SSH: Sparse Spectrum Adaptation via Discrete Hartley Transformation. NAACL 2024.

- A sensivity analysis on $r$ value, in conjunction with different methods, should be necessary to show its robustness.

- More experimental justification is needed on whether the proposed method is dependent on specific ViT shapes (or not).

- More implementation details, for example, on how $U$, $V$ are materialized and shared, should be made.

**Questions:**

Please refer to the weakness section, and address the concerns point-by-point.

---

> ### Author Response · Authors · 2025-12-03
> **Response to Reviewer Comments**
>
> We thank the reviewer for the constructive feedback. Below we respond point by point.
>
> ------
>
> ### **Q1. Theoretical justification and subspace invariance**
>
> CERSA **does not** assume individual singular vectors remain fixed. It assumes only that the **principal subspace** is stable under fine-tuning—a weaker and empirically validated condition. Our analysis in Sec. 3.3 shows that if this subspace is preserved, the fine-tuned weight can be written as
> $$
> W' \approx U_p S_p V_p^\top,
> $$
> meaning that learning only $S_p$ is equivalent to allowing arbitrary rotations and rescalings **within** the subspace, preserving expressiveness while reducing memory.
>
> Empirically, we observe extremely high subspace stability:
>
> - Appendix. Table 8: Across eight datasets, Q/K/V subspace similarity for ViT-Large remains **99%–99.99%** under full fine-tuning.
> - Additional analyses (see Q2) show this still holds under distribution shift.
>
> Thus, CERSA does not rely on fixed directions; it leverages the observed stability of the **span**, while $S_p$ flexibly adapts within it.
>
> ------
>
> ### **Q2. Behaviour under distribution shift (CIFAR-100-C)**
>
> To evaluate robustness under **input-level** shift, we fine-tuned on CIFAR-100-C (Gaussian Noise, severity 3). Subspace similarity remains extremely high:
>
> | Matrix | CIFAR-100   | CIFAR-100-C     |
> | ------ | ----------- | --------------- |
> | Q      | 99.69/99.65 | **99.69/99.63** |
> | K      | 99.76/99.74 | **99.73/99.70** |
> | V      | 99.58/99.58 | **99.58/99.59** |
>
> The change relative to clean data is <0.03%, confirming fine-tuning continues to operate within the original principal subspace.
>
> Performance under corruption shows the same pattern:
>
> | Method           | CIFAR-100-C (Gaussian Noise) |
> | ---------------- | ---------------------------- |
> | Full FT          | 93.6                         |
> | LoRA             | 87.4                         |
> | **CERSA (0.95)** | **93.4**                     |
>
> CERSA nearly matches full fine-tuning while LoRA degrades strongly. These results, combined with our OOD experiments, indicate that **even under corruption, the principal subspace remains a robust basis for adaptation**. We will move a subset of these results into the main text.
>
> ------
>
> ### **Q3. Missing recent PEFT baselines**
>
> We thank the reviewer for pointing out these very recent PEFT methods. To directly address this concern, we compared CERSA with **VeRA**, **FourierFT** and **SSH** on the datasets that are jointly reported in the SSH paper (CIFAR-100, DTD, EuroSAT, OxfordPets). The results are:
>
> | Method    | CIFAR-100 | DTD      | EuroSAT  | OxfordPets | Avg.     |
> | --------- | --------- | -------- | -------- | ---------- | -------- |
> | VeRA      | 94.2      | 81.6     | 98.6     | 93.7       | 92.0     |
> | FourierFT | 93.7      | 81.2     | 98.7     | 94.5       | 92.0     |
> | SSH       | 94.5      | 81.9     | 99.0     | 94.8       | 92.6     |
> | **CERSA** | **94.3**  | **82.5** | **99.1** | **94.9**   | **92.7** |
>
> These results confirm that our conclusions are not an artifact of missing baselines: CERSA remains competitive or superior even against the latest PEFT methods specifically designed for strong task performance.  Note that VeRA/SSH operate on top of full frozen weights, whereas CERSA also compresses the pre-trained weights themselves, leading to lower total memory under the same setting.
>
> ------
>
> ### **Q4. Sensitivity of $\alpha$ and $\beta$**
>
> Appendix Table 2 shows CERSA is robust across a wide range of hyperparameters:
>
> - With $\alpha=\beta$ from **0.95 to 0.8**, accuracy drops only **1.4%** while memory decreases significantly.
> - Fixing $\alpha=0.95$  and reducing β from **0.95 to 0.8** changes accuracy by at most **0.5%**; even $\beta$ = 0.5 yields only ~3.2% drop.
>
> Thus, CERSA maintains stable performance across broad settings. We will include a summarized plot in the main paper.
>
> ------
>
> ### **Q5. Dependence on ViT shape**
>
> CERSA is **not** specialized to a specific ViT shape. Results on ViT-Base and ViT-Large across eight datasets show consistent trends:
>
> - ViT-Base: highest average accuracy among PEFT baselines.
> - ViT-Large: best accuracy–memory trade-off (Fig. 1–2).
>
> Fig. 3 further shows that retained-rank patterns are structurally similar across Q/K/V and MLP layers, indicating dependence on **spectral statistics**, not specific shape.
>
> ------
>
> ### **Q6. Implementation details of $U_p, S_p, V_p$**
>
> We clarify the implementation:
>
> - Each weight matrix undergoes one SVD at initialization. We keep truncated $U_p$ and $V_p$ as **frozen buffers**.
> - We train only $S_p$, initialized with top-k singular values on the diagonal.
> - Each layer uses its own rank and factors; no cross-layer parameter sharing.
> - Memory accounting in Appendix C includes weights, gradients, and optimizer states of $S_p$. Despite the $k^2$ trainable parameters, total memory remains lower than LoRA at comparable accuracy.
>
> These clarifications will be added to Sec. 4 and Appendix B.

---

### Note · Authors · 2026-01-26

I have read and agree with the venue's withdrawal policy on behalf of myself and my co-authors.

---

### Meta-Review · Area_Chair_kDSL · 2026-01-06

**Summary:**

This manuscript argues that in fine-tuning scenarios, storing the frozen pretrained weights remains necessary, which limits efficiency. To address this issue, the authors propose Cumulative Energy-Retaining Subspace Adaptation (CERSA). The basic idea is to on the one hand cutting the pre-trained weights to a low-rank representation by SVD and on the other hand training only a core matrix.

The reviewers’ opinions on this work are mixed. Two reviewers consider the idea novel, while Reviewer 39qq views the main contribution as incremental, and Reviewer BtA2 argues that the method is closely related to existing SVD-based PEFT approaches. The AC shares a similar concern: the proposed method can be seen as an extension of the idea of training a core matrix for LoRA to the pretrained weights. However, this raises a conceptual issue. While it is generally accepted that weight updates are often low-rank, the pretrained weights themselves do not necessarily possess a low-rank structure. This concern is also related to Reviewer iRHp’s point that the update geometry and useful directions are typically not fixed to a predefined low-rank subspace.

The experimental evaluation was questioned by reviewers. In the AC’s view, the authors’ rebuttal addressed some of these concerns; unfortunately, the reviewers did not have the opportunity to respond to the newly added results.

The authors further argued that one reviewer with a negative score used AI-generated text. It is important to clarify that neither the AC nor anyone else is in a position to judge whether a review was AI-generated. Moreover, reviewer comments represent the reviewers’ opinions regardless of how they were written. The AC’s role is limited to assessing whether the comments are logical and relevant, not their origin.

Overall, the paper contains potentially interesting ideas, and the reported performance appears promising. Unfortunately, many of the stronger experimental results were only introduced during the rebuttal (were not evaluated by the reviewers) and may show that the intial submission has improvement potentials. More importantly, several key conceptual issues remain insufficiently explained. Overall, for its current version, the AC recommends rejection.

**Reviewer Concerns:**

This paper received four reviews, with two reviewers expressing positive opinions and the other two holding negative ones. Among the negative reviews, one primarily focused on experimental shortcomings, while the other raised multiple conceptual and methodological questions.

Regarding the method itself, Reviewers 39qq and BtA2 considered the core innovation relatively incremental, while Reviewer iRHp questioned the appropriateness of using a fixed low-rank subspace.

Regarding the experimental evaluation, Reviewer R2ux appreciated the broad range of experiments (especially including catastrophic forgetting). However, Reviewer BtA2 felt the evaluation scale was limited, and Reviewers 39qq and iRHp noted the absence of comparisons with strong LoRA baselines.

The authors did provide additional experiments during the rebuttal. After examining these results, the AC believes that some concerns were partially addressed. However, the omission of several important and feasible experiments in the original submission suggests that the paper was either not fully prepared or could still be substantially improved.

**Reviewer Scores:**

The initial score is 2/4/6/6. Unfortunately, there is no response received from the reviewers.

---

### Decision · Program_Chairs · 2026-01-26

Reject